# MedBLINK: Probing Basic Perception in Multimodal Language Models for Medicine

## Abstract

Multimodal language models (MLMs) show promise for clinical decision support and diagnostic reasoning, raising the prospect of end-to-end automated medical image interpretation. However, clinicians are highly selective in adopting AI tools; a model that makes errors on seemingly simple perception tasks such as determining image orientation or identifying whether a CT scan is contrast-enhanced, are unlikely to be adopted for clinical tasks. We introduce MedBLINK, a benchmark designed to probe these models for such perceptual abilities. MedBLINK spans eight clinically meaningful tasks across multiple imaging modalities and anatomical regions, totaling 1,429 multiple-choice questions over 1,605 images. We evaluate 20 state-of-the-art MLMs, including general-purpose (GPT-5, Gemini 1.5 Pro, Claude 3.5 Sonnet) and domain-specific (Med-Flamingo, LLaVA-Med, RadFM) models. While human annotators achieve 96.4% accuracy, the best-performing model reaches only 76.3%. These results show that current MLMs frequently fail at routine perceptual checks, suggesting the need to strengthen their visual grounding to support clinical adoption.

## 1 Introduction

Would you trust ChatGPT if it failed to identify whether an image was upside down? For artificial intelligence (AI) systems to be adopted, they must demonstrate competence not only on complex benchmarks, but also on simple, intuitive tasks. The same expectation holds perhaps even more critically for AI in medicine, where failures on basic perceptual cues can erode clinician confidence. As multimodal language models (MLMs) increasingly enter clinical settings (Tu et al., 2023; McDuff et al., 2023; Moor et al., 2023), their reliability on foundational tasks becomes as important as their performance on advanced diagnostic reasoning.

Recent advances in vision-language modeling have sparked enthusiasm about fully automated systems that can interpret medical images and inform clinical decision-making (Yang et al., 2024b; Lu et al., 2024; Li et al., 2023). Yet clinicians remain appropriately cautious in adopting AI tools (Bedoya et al., 2019; Guidi et al., 2015; Tonekaboni et al., 2019). Physicians rely on deeply internalized mental models for interpreting medical images, and they expect AI to match this fluency. Models that fail on what clinicians consider "obvious" tasks like detecting image orientation or identifying contrast-enhancement, risk immediate dismissal, regardless of downstream capabilities (Asan et al., 2020). For example, knowing whether a CT scan is contrast-enhanced directly influences downstream diagnostic interpretation and subsequent clinical decisions (Inamdar & Shinde, 2024).

These basic assessments, often termed "blink tasks"(Fu et al., 2024), occur almost reflexively in expert workflows. They rely on low-effort perceptual and contextual cues, not elaborate reasoning or cross-modal fusion. If a model struggles here, it signals a failure to internalize visual priors critical for real-world use (Morita et al., 2008). It raises questions about whether the model genuinely "sees" the content of medical images or simply exploits superficial correlations (Sepehri et al., 2024).

We introduce MedBLINK, a benchmark designed to probe exactly these capabilities. MedBLINK comprises eight perceptually simple but clinically meaningful tasks chosen by consulting with a senior radiologist. The questions are deliberately simple, asking models to perform basic visual perception rather than complex reasoning. Fig. 1 illustrates one example from each task, including cases like visual depth estimation and image orientation detection. Failures on these tasks would

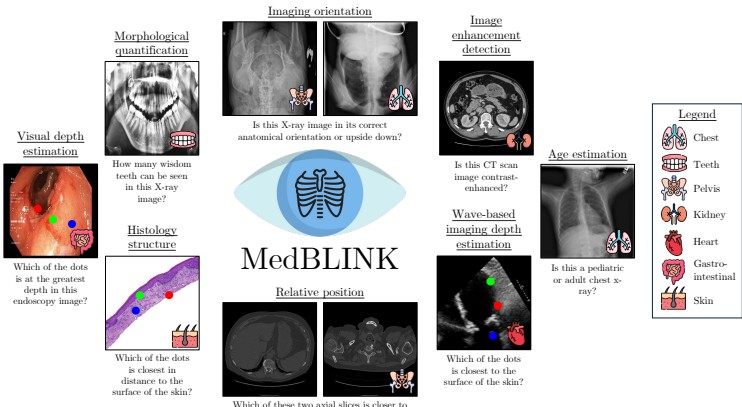

Figure 1: The visual tasks that medical professionals could solve within a blink but MLMs struggle. These tasks cover a range of clinically relevant problems, including anatomical orientation, morphology qualification, visual and wave-based depth estimation, and histology analysis.

reveal that models struggle to capture spatial relationships and maintain a coherent understanding of anatomical structures.

We evaluate 20 state-of-the-art MLMs, including general-purpose models such as GPT-5 (OpenAI, 2025), GPT-4o (Hurst et al., 2024) and Claude 3.5 Sonnet (cla, 2024), as well as medical-domain models such as Med-Flamingo (Moor et al., 2023), LLaVA-Med (Li et al., 2023), and RadFM (Wu et al., 2023b). While human annotators achieve 96.4% accuracy, the best-performing model reaches only 76.3%. By probing what models fail to perceive, not just what they fail to predict, MEDBLINK highlights a fundamental gap in current evaluation protocols. Addressing this gap is essential: models must first master the same low-effort, common-sense perceptual tasks before they can be trusted to support real-world diagnostic reasoning and clinical adoption.

## 2 RELATED WORK

**Multimodal Language Models in Healthcare**: Medical image analysis has evolved from early computer-aided detection efforts (Lodwick et al., 1963; Giger et al., 2008; Winsberg et al., 1967) to recent advances in Multimodal Language Models (MLMs) (Touvron et al., 2023; Alayrac et al., 2022; Radford et al., 2021; Li et al., 2024a). These models combine text and image understanding and are typically evaluated using visual question answering (VQA) tasks. Their adoption in healthcare has enabled progress on diagnostic and report generation tasks across various domains (Tu et al., 2023; Yang et al., 2024b; Moor et al., 2023; Wang et al., 2025; Li et al., 2023; Cui et al., 2024; Hurst et al., 2024). However, limitations persist due to the lack of large-scale multimodal medical datasets and the heterogeneity of medical image formats (e.g., 2D X-rays, 3D CT/MRI, video, gigapixel histology). This has motivated domain-specific models such as VoxelPrompt (Hoopes et al., 2024) and Dia-LLaMA (Chen et al., 2024b) for volumetric imaging, and Quilt-LLaVA (Seyfioglu et al., 2024), PathChat (Lu et al., 2024), and PathFinder (Ghezloo et al., 2025) for histopathology.

**Multimodal Benchmarks in Medicine**: Growing medical MLM development has spurred numerous benchmarks evaluating performance across modalities and tasks, primarily assessing medical knowledge (Lau et al., 2018; Liu et al., 2021; 2024; Wu et al., 2023a; He et al., 2020; Hu et al., 2024; Sepehri et al., 2024; Royer et al., 2024; Zhang et al., 2023; Wu et al., 2023b; Ye et al., 2024; Seyfioglu et al., 2024; Xia et al., 2024). SLAKE (Liu et al., 2021) and VQA-RAD (Lau et al., 2018) sample radiology images, from existing datasets, to create clinical diagnostic QA pairs. Path-VQA (He et al., 2020) curates pathology images from textbooks with QA pairs from captions. Quilt-VQA (Seyfioglu et al., 2024) benchmarks histopathology from pedagogical videos, extracting QA from transcriptions. OmniMedVQA (Hu et al., 2024) develops large-scale VQA covering 12 medical modalities. GMAI-MMBench (Ye et al., 2024) leverages 38 modalities for perceptual tasks beyond diagnosis. CARES (Xia et al., 2024) assesses trustworthiness across 16 modalities in five dimensions: trustfulness, fairness, safety, privacy, robustness. MediConfusion (Sepehri et al., 2024) probes failure

modes on visually dissimilar image pairs. RadVUQA (Nan et al., 2024) highlights critical gaps in spatial, anatomic, and quantitative reasoning. Unlike complex deductive benchmarks, BLINK (Fu et al., 2024) shows perceptually demanding tasks remain challenging for MLMs. MEDBLINK extends this line of work by targeting foundational perceptual skills that are easy for clinicians but consistently missed by current models. Rather than emphasizing complex reasoning, it probes the basic visual competencies essential for earning trust in clinical deployment—filling a critical gap in how model trustworthiness is currently evaluated.

## 3 MEDBLINK BENCHMARK

Clinical image interpretation relies on both perceptual and conceptual reasoning. Perception enables clinicians to extract key visual features before engaging in more complex diagnostic inference (Morita et al., 2008). Yet most medical AI benchmarks focus only on conceptual tasks, assuming that strong diagnostic performance implies adequate visual understanding. This creates a critical blind spot: models may generate plausible answers without genuinely perceiving the image.

MEDBLINK evaluates this foundational trust layer by testing MLMs on perceptually simple yet clinically important tasks such as counting, depth estimation, anatomical orientation, and enhancement detection. These reflect early visual judgments clinicians perform routinely; poor performance signals perceptual grounding gaps undermining downstream trust.

The benchmark includes 1,429 multiple-choice questions over 1,605 expert-validated images across diverse modalities and anatomical regions, reusing and augmenting existing datasets with single or paired image tasks (Tab. 7, Sec. 3.2). All samples underwent manual review for clarity, quality, and ambiguity, with domain expert feedback guiding refinement. MEDBLINK serves as both a perceptual benchmark and a focused probe of model trustworthiness in clinical settings.

### 3.1 MEDBLINK CHARACTERISTICS AND FEATURES

MEDBLINK covers 5 ubiquitous medical modalities including: X-ray, CT, Endoscopy, Histopathology, and Ultrasound, and measures performance across multiple anatomical regions, including dental, chest, skin, pelvic, abdominal, cardiac, renal, and gastrointestinal systems.

The modalities are selected to cover a wide range of image acquisition methods (e.g. X-ray, CT, Ultrasound, Endoscopy, and Histopathology), output type/dimensions (e.g. 3D CT scans or histopathology giga-pixel images), and anatomical regions. Hence, improved performance on MEDBLINK suggests improvements in the broader set of medical imaging modalities (e.g., CT is similar to MRI and PET in its 3D dimensions, Fluoroscopy utilizes similar radiation acquisition methods as X-ray, and Ultrasound has a similar rigid structure to OCT).

MEDBLINK features key novelties when compared with other medical benchmarks:

**Perceptual Tasks**: In contrast to other medical multimodal benchmarks, we explore medical visual tasks which are seemingly simple albeit clinically significant tasks, essential for ensuring accurate diagnoses and decision.

**Diverse and Generalizable Tasks**: Our data is sourced from diverse imaging modalities and anatomic areas, and our tasks are generalizable to other modalities not covered in this benchmark.

**Visual Prompting**: Clinicians often focus on specific image regions, when reviewing or communicating findings. We mimic this form of prompting, by leveraging visual cues like points/dots to spatially prompt the models when answering the specified question (Seyfioglu et al., 2024; Nan et al., 2024).

**In-domain imaging properties**: We explicitly test models on the clinical characteristics of medical imaging including knowledge of structural asymmetries, anatomic geometric reasoning, clinical relative depth estimation, and quantification of features leveraging morphology.

### 3.2 MEDBLINK CURATION

**Task 1: Image Enhancement Detection** Image enhancements in medical imaging, such as contrast injections, improve quality and highlight important structures when unenhanced scans are insufficient. This task tests whether models can distinguish between enhanced and unenhanced CT images. We use the VindDr Multiphase dataset (Dao et al., 2022), which contains Non-Contrast, Arterial, and Venous phase CT scans. We extract abdominal slices that include the kidneys manually, where contrast

effects are clearly visible. Each image is paired with a question asking whether it is enhanced; see Tab. 9 for sample prompts.

**Task 2: Visual Depth Estimation** This task evaluates a model's ability to infer relative depth in medical images captured by RGB-based modalities such as endoscopy. These imaging techniques produce 2D representations of 3D anatomical structures, which experts can intuitively interpret by mapping pixel positions to spatial depth, in the same way one can look at an image and deduce relative depth (Chen et al., 2016).

To test this capability, we use endoscopic images from the Kvasir dataset (Pogorelov et al., 2017). For each question, we present a randomly selected image with a bounding box at the center (covering 25% of the image area). Inside the box, three colored points are placed at varying depth levels, determined using depth maps generated by the DepthAnything model (Yang et al., 2024a) and subsequently validated.

**Task 3: Wave-Based Imaging Depth Estimation** Similarly, this task tests the performance of models on estimating the relative depth of spatial regions in medical modalities that leverage wave-based acquisition techniques e.g. Ultrasound with sound waves, or OCT with light waves. The underlying physical principles of acoustic wave propagation dictate that ultrasound presents as a characteristic cone-like structure (Szabo, 2013) with the highest visible features in the cone closest to the point of contact with the skin.

The objective of this task is to evaluate MLMs understanding of depth in wave-based medical imaging modalities. We do this by placing visual points within the cone and asking which point is closest to the point of contact. We employ the EchoNet-Dynamic echocardiogram (Ouyang et al., 2020) video dataset. We select 150 videos and randomly select one frame from each of the 150 videos. Then we use grayscale thresholding to create a mask of the ultrasound cone and divide the mask three thirds, with a 10-px gap between each third, and place one dot randomly in each third, assigning a random color (red, blue, green) to each point with the point/dot in the top third is the closest to the skin/point-of-contact. To evaluate the MLMs, we randomly flip the images uniformly across 4 orientations: 0-degree (upright), 90-degree, 180-degree (upside down), and 270-degree clockwise rotation.

**Task 4: Histology Structure** With this task, we evaluate how well models understand the non-rigid structures of medical images. Unlike other medical domains like X-ray images with rigid structures outlined by the human skeleton, many histopathology subdomains do not have any strict anatomy structure as the imaged tissue samples are extracted from suspected cancerous tissue, however, certain sub-domains like skin histology maintain a visible structure of cell layers: epidermis and dermis (Arda et al., 2014) with the dermis underlying the epidermis structurally.

We evaluate models basic knowledge of these layers by placing visible points (using contrasting colors) in the epidermis, and dermis by asking which point is closest to the surface of the skin. We leverage the dataset of Hematoxylin and Eosin-stained (H&E) skin whole slide images (WSI) (Salam et al., 2025) and the provided masks, segmenting 12 tissues classes including skin layers of epidermis and dermis. We crop representative images and randomly place points/dots in the epidermis and dermis areas to curate our images for the task, while noting the color of the correct dot.

**Task 5: Imaging Orientation** Here, we benchmark models on identifying incorrectly oriented medical images of modalities with strong structural priors, for example, is an x-ray image upside down or not? The correct spatial orientation of medical images is important to accurately interpret anatomical structures; inherent structural asymmetries within human physiology serve as orientation landmarks in images, such as the cardiac silhouette's leftward projection, and the liver's right-sided dominance in X-ray images. Human experts can intuitively recognize these landmarks and reorient misaligned images based on anatomical priors.

For this task, we evaluate the ability of multimodal LMs to identify incorrectly oriented medical images. To construct the task we leverage the test split of ChestX-ray8 (Wang et al., 2017) adult (age: > 20) chest x-ray dataset and a pelvic x-ray dataset [1]. We randomly sample 200 patient images from each dataset, and we randomly flip (180-degree) 100 of the samples and ask if the image is correctly oriented or not.

---

[1] github.com/yaufan/Pelvis-X-ray_Segmentation_Database

**Task 6: Relative Position** This task tests whether models understand 3D human anatomy by asking them to determine which 2D slice is closer to specific organs. 3D medical modalities like CT and MRI fundamentally take a snapshot of the fixed internal structure of the human anatomy; therefore, given any slice of a CT scan, it is relatively trivial to deduce the position relative to other slices e.g. given two axial slices from the chest and abdomen, one can tell which slice is closer to specific organs.

We test the 3D anatomy mental models of the MLMs to deduce if they memorize seen samples or fundamentally understand the 3D structure of human scans. To do so, we leverage the OSIC Pulmonary Fibrosis Progression (Al Nazi et al., 2021)[2] dataset which has 176 CT scans to curate visually separable slices and ask which of the slices is closer to a fixed organ. For this, we segment the slices along the depth dimension into 3 bins and subsequently we randomly sample two images: one from the first bin and the other from the last bin, to have reasonable visual separation in the content of the slices i.e. typically the first slice is from the shoulder/chest region and the latter from the chest/abdomen area. The images are concatenated side-by-side and labeled (1 or 2) and used to task the MLMs with predicting which image is closest to the Pelvis. To break from expectation, we randomly shuffle which image is first in the collage.

**Task 7: Morphology Quantification** Here, we test if models can count important medical features (e.g., cells) in medical images based on the features morphology. In medical imaging, a significant amount of clinical scores that determine diagnosis and subsequently patient care are based on counting the occurrence of certain features. For example, in radiology, many stratification scoring systems also depend on counting visible features, e.g. counting the number of nodules in CT images provide additional context for estimating Lung-RADS score (Martin et al., 2017).

To make sure that the vision encoder's patch constraints do not factor in the MLM's performance on counting accurately, we leverage the Panoramic dental radiography (Budagam et al., 2024) dataset with the masks to count the number of wisdom teeth (3rd molars) in the radiographs given 3 options, as they are visually more prominent than, for example, cells in WSI, and typically have no occlusions.

**Task 8: Age Estimation** This task evaluates the ability of Multimodal LMs to estimate the age group of patients based on solely on the clinical presentation in chest X-ray images. This task require the identification of anatomical differences present in images, for example, pediatric patients exhibit a proportionally larger heart compared to the adults, and the thoracic cage in children appears more circular with horizontally oriented ribs, in contrast to the elliptical cage with oblique ribs seen in adults (Brakohiapa et al., 2017). For the task we collect 100 pediatric (age: $< 7$) and 100 adult (age: $> 20$) unique patient chest x-ray from the train split of the ChestX-ray8 (Wang et al., 2017) dataset.

## 4 EXPERIMENTS

We evaluate 20 state-of-the-art MLMs on MEDBLINK. First, we find that while human performance is consistently high, current models struggle significantly on medical visual perception tasks, particularly on enhancement detection and counting tasks. Second, we find that scaling model parameters improves performance across most tasks with diminishing returns. Third, we find that medical MLMs perform the worst relative to other baselines, despite in-domain training, and we discuss why. Lastly, while API-based models perform well on general domain tasks like orientation detection, they perform poorly on medical orientation, signaling a poor medical perceptual understanding distinct from general perceptual understanding.

### 4.1 MODELS EVALUATED

We evaluate MEDBLINK on 20 current Multimodal LMs across three groups: **Medical Multimodal LMs:** We measure the performance of 4 medical domain-specific models trained on medical data: LLaVA-Med (Li et al., 2023), Med-Flamingo (Moor et al., 2023), LLAVA-MED++ (Xie et al., 2024) and RadFM (Wu et al., 2023b). **Open-Source Multimodal LMs:** We evaluate 9 general-purpose open-source models on all tasks: Qwen 2.5 VL (3B and 7B) (Bai et al., 2025), INTERNVL 2.5 (4B, 8B, 26B, 38B) (Chen et al., 2024a) and LLaVA-OneVision (0.5B and 7B) (Li et al., 2024b), LLAMA 3.2 11B (Grattafiori et al., 2024) and 3 spatial-specialized models on the tasks that require

---
[2]kaggle.com/datasets/donkeys/osic-pulmonary-fibrosispreprocessed

Table 1: **Results of different models on MEDBLINK.** All values represent accuracy (%) on each task. The first row shows abbreviated task names along with the number of test samples. The best model performance for each task is underlined.

| | CONTRAST (134) | EST. AGE (200) | ORIENT. CXR\|PV (200\|200) | HISTO. ST. (141) | REL. POS. (176) | WAVE DEPTH (146) | VIS. DEPTH (144) | QUANT. FTS. (88) | Average (1429) |
|---|---|---|---|---|---|---|---|---|---|
| Random Choice | 50.0 | 50.0 | 50.0\|50.0 | 33.3 | 50.0 | 33.3 | 33.3 | 33.3 | 42.58 |
| Experts | **97.5** | 93.6 | **100.0\|100.0** | **99.2** | **100.0** | **100.0** | 95.1 | **81.8** | **96.36** |
| Specialized models | - | 98.5 | 100.0\|100.0 | - | - | - | - | - | - |
| **Medical Multimodal LLMs** | | | | | | | | | |
| LLAVA-MED (Li et al., 2023) | 50.0 | 50.0 | 50.0\|50.0 | 39.0 | 57.4 | 34.2 | 47.9 | 14.7 | 43.69 |
| RADFM (Wu et al., 2023b) | 50.0 | 63.0 | 60.5\|48.0 | 20.6 | 69.3 | 32.9 | 26.4 | 34.1 | 44.98 |
| MED-FLAMINGO (Moor et al., 2023) | 50.0 | 81.5 | 50.0\|50.0 | 29.1 | 72.1 | 33.6 | 29.1 | 31.8 | 47.47 |
| LLAVA-MED++ (Xie et al., 2024) | 50.0 | 92.5 | 50.0\|50.0 | 41.4 | 34.7 | 31.5 | 35.4 | 34.1 | 46.62 |
| **Open-Source Multimodal LLMs** | | | | | | | | | |
| LLAVA-ONEVISION (0.5B) (Li et al., 2024b) | 50.0 | 68.5 | 50.0\|50.0 | 36.1 | 27.8 | 33.5 | 43.7 | 34.1 | 43.74 |
| LLAVA-ONEVISION (7B) (Li et al., 2024b) | 50.0 | 84.0 | 59.0\|69.0 | 41.1 | 20.5 | 43.1 | 54.2 | 30.7 | 50.18 |
| QWEN 2.5 VL (3B) (Bai et al., 2025) | 50.7 | 64.5 | 50.0\|50.0 | 38.3 | 22.7 | 34.2 | 43.0 | 32.9 | 42.9 |
| QWEN 2.5 VL (7B) (Bai et al., 2025) | 50.0 | 87.0 | 50.5\|50.0 | 43.2 | 28.4 | 36.3 | 61.1 | 37.5 | 49.33 |
| INTERNVL 2.5 (4B) (Chen et al., 2024a) | 50.0 | 72.5 | 50.5\|50.0 | 36.2 | 14.7 | 36.3 | 40.3 | 35.2 | 42.86 |
| INTERNVL 2.5 (8B) (Chen et al., 2024a) | 50.0 | 87.0 | 50.0\|50.0 | 37.6 | 10.8 | 37.7 | 67.3 | 30.8 | 46.80 |
| INTERNVL 2.5 (26B) (Chen et al., 2024a) | 50.0 | 80.5 | 50.0\|50.0 | 46.8 | 17.6 | 41.8 | 65.3 | 37.5 | 48.83 |
| INTERNVL 2.5 (38B) (Chen et al., 2024a) | 50.0 | 74.0 | 50.0\|50.0 | 46.1 | 19.3 | 52.1 | 65.3 | 34.1 | 48.99 |
| LLAMA 3.2 11B (Grattafiori et al., 2024) | 50.0 | 86.0 | 49.5\|50.0 | 59.6 | 44.9 | 33.6 | 45.8 | 39.8 | 51.02 |
| **API-Based Models** | | | | | | | | | |
| CLAUDE 3.5 SONNET (cla, 2024) | 50.0 | 89.0 | 82.4\|100.0 | 52.2 | 61.0 | 38.2 | 73.5 | 38.6 | 64.99 |
| GEMINI 1.5 PRO (Team et al., 2024) | 57.1 | 94.7 | 93.3\|79.5 | 64.0 | 21.5 | 60.6 | 76.5 | 35.6 | 64.76 |
| GPT-4O (Hurst et al., 2024) | 50.3 | 86.7 | 80.1\|67.6 | 44.3 | 43.7 | 42.9 | 67.4 | 12.9 | 55.1 |
| GPT-5 (OpenAI, 2025) | 79.8 | 95 | 91.2\|98.7 | 51.1 | 81.8 | 46.6 | 91 | 51.7 | 76.3 |
| **Ablation** | | | | | | | | | |
| CLAUDE 3.5 SONNET fewshot | 84.1 | 86.5 | 94.7\|100.0 | 56.5 | 73.9 | 56.1 | 81.8 | 51.4 | 76.1 |
| CLAUDE 3.5 SONNET zeroshot | 50.0 | 90.4 | 80\|100.0 | 54.6 | 59 | 43.1 | 74.5 | 40 | 65.7 |
| LLAVA-ONEVISION fewshot | 50 | 50.5 | 52.5\|50 | 31.2 | 29.9 | 42 | 46.1 | 22.3 | 41.6 |
| LLAVA-ONEVISION zeroshot | 50.0 | 83.8 | 59.1\|69.2 | 40.6 | 20.7 | 43.3 | 53.2 | 31.8 | 50.2 |
| LLAVA-MED++ (Freeform) (Xie et al., 2024) | 50.0 | 94.0 | 50.0\|50.0 | 35.5 | 27.8 | 32.2 | 27.1 | 30.7 | 44.14 |

spatial depth reasoning: AURORA (Bigverdi et al., 2024), SpatialRGPT (Cheng et al., 2024), and LLaVA 1.5 (7B) (Liu et al., 2023). **API-based Multimodal LMs:** We test 4 proprietary models: GPT-5 (OpenAI, 2025), GPT-4o (Hurst et al., 2024), Claude 3.5 Sonnet (cla, 2024), and Gemini 1.5 Pro (Team et al., 2024). Finally, we also benchmark small specialized CNN models trained with ResNet-50 (He et al., 2015) on the training sets of the underlying datasets used to construct MEDBLINK, see Sec. B.1.

## 4.2 EXPERIMENTAL PROCEDURE

We follow BLINK's evaluation setup (Fu et al., 2024; Duan et al., 2024), setting temperature to 0, adjusting retries to 5 (2 for GPT-5), and not resizing images. For uniformity, we concatenate images for multi-image tasks (e.g., 3D relative positioning on CT). We leverage clinical experts for human evaluation, use uniform visual prompt sizing based on image dimensions, and report model accuracy. See appendix Sec. A for details.

## 4.3 RESULTS

**Multimodal Models Remain Far from Trustworthy Performance.** While human experts achieve 96.36% average accuracy across the benchmark, even the best-performing model (GPT-5) achieves only 76.3% accuracy. As shown in Tab. 1, API-based models perform best (55.1-76.3%), followed by open-source models (42.86-51.02%), with medical domain-specific models surprisingly performing worst (43.69-47.47%) despite their specialized training.

**Models struggle most with perceptual reasoning tasks requiring contrast detection and counting.** On the contrast identification task, most models perform at random chance, with API-based models averaging only 59.3%, well below the 97.5% achieved by human experts. Morphology quantification proves even more challenging: all models perform poorly, with GPT-4O achieving just 12.9% accuracy compared to random chance at 33.3% and human performance at 81.8%. These results underscore fundamental limitations in MLMs' ability to perceive fine-grained visual differences in medical images. Notably, even GPT-5, despite leading overall with 76.3% accuracy, exhibits substantial weaknesses on these tasks.

**Medical-specific MLMs underperform general models despite domain specialization.** Counterintuitively, domain-specific medical models achieve lower average performance (45.7%) than API-based (65.3%) and open-source (47.2%) models. LLAVA-MED performs at random chance or

below on five out of eight tasks and struggles particularly on morphology quantification (14.7%). RADFM and MED-FLAMINGO show notably weak performance on histology structure (20.6% and 29.1%, respectively) and visual depth estimation (26.4% and 29.1%), suggesting these specialized models may develop spurious correlations on diagnostic tasks rather than meaningful medical perceptual understanding towards answering complex diagnostic questions. While LLAVA-MED++ outperforms other medical models on EST. AGE task with 92.5%, it underperforms on the REL. POS. task and on average with 46.62% despite its larger pretraining-data size.

**Larger models consistently outperform smaller variants on most tasks.** As shown in Tab. 1, LLaVA-OneVision 7B outperforms its 0.5B counterpart on average (50.18% vs. 43.74%), Qwen 7B exceeds Qwen 3B (49.33% vs. 42.9%). Parameter scaling on INTERNVL 2.5 showed the same trend with scale: 3.94% improvement from 4B to 8B, 2.03% from 8B to 26B, and only 0.16% from 26B to 38B on average. This scaling effect is particularly pronounced in tasks like estimating age from chest X-ray (84.0% vs. 68.5% for LLaVA-OneVision) and visual depth estimation (61.1% vs. 43.0% for Qwen), suggesting that increased parameter count benefits complex perceptual reasoning tasks. However, this trend occasionally reverses for specific tasks, such as relative positioning where LLaVA-OneVision 0.5B outperforms the 7B variant (27.8% vs. 20.5%).

**Specialist CNNs Easily Solve MEDBLINK Tasks.** To assess the inherent difficulty of certain tasks in MEDBLINK, we trained ResNet-50 models on the training sets corresponding to three tasks: EST. AGE, chest X-ray orientation, and pelvic X-ray orientation. When evaluated on the corresponding test sets within MEDBLINK, the models achieved 98.5% accuracy on EST. AGE and 100% on both orientation tasks. These results suggest that these tasks are perceptually simple—easy enough to be solved reliably by small convolutional neural networks, highlighting that current MLM failures stem from limitations in visual grounding rather than task complexity.

**Models frequently resort to heuristics instead of accurate perception.** In the imaging orientation task, RADFM incorrectly identified most flipped chest X-rays as being correctly oriented, demonstrating poor understanding of basic anatomical orientation. Similarly, in histology structure tasks, MED-FLAMINGO and RADFM frequently defaulted to predicting the blue dot as closest to the surface of the skin regardless of the actual position of the dots in the various tissue layers. In the wave-based depth estimation task, medical models like MED-FLAMINGO, LLAVA-MED, and RADFM, as well as, general-purpose models like GEMINI 1.5 PRO and CLAUDE 3.5 SONNET, frequently defaulted to predicting the red dot as the answer. This suggests reliance on color-based heuristics or spurious correlations rather than accurately perceiving depth information in medical images. Our analysis reveals recurring failure patterns across models on MEDBLINK tasks, see Fig. 2 below.

**Models fail at depth and contrast perception despite confident reasoning.** Some models, including GEMINI 1.5 PRO, and CLAUDE 3.5 SONNET provide explanations/textual-reasoning for their prediction. On the visual depth estimation task with endoscopic images, these explanations do not reflect the actual depth relationships within the images, see Fig. 4. On the enhancement detection tasks leveraging CT slices, models fail to classify contrast-enhanced CT images with incorrect explanations, see Fig. 12, which suggests that they may be failing to detect important anatomical features for CT enhancement, such as the aorta, that human experts use when performing these tasks.

## 4.4 ABLATION STUDIES

**Can Models Perform Spatial Reasoning in the Medical Domain?** While recent models have shown improved spatial reasoning capabilities on natural domain images, our results in Tab. 2 demonstrate that these advances do not translate to medical imaging. Models specifically developed for improved spatial reasoning, like SpatialRGPT only achieve 41.3% average accuracy across tasks that require visual prompts —barely above random chance (33.3%). Similarly, AURORA and LLaVA1.5-7B perform near random levels despite their capabilities on natural images. Notably, while Gemini shows better performance (67.0% average), it still falls far short of expert-level accuracy (98.1%). These findings suggest that spatial reasoning in medical imaging presents unique challenges beyond those in natural domains.

**Do Prompting Strategies Actually Help?** To investigate prompting strategy impact, we conducted experiments using four approaches on LLAVA-MED++: Freeform (FF), OmnimedVQA's (Hu et al., 2024) Question-answering (OQA), Prefix-based (OPB), and Multiple-choice (MC) across ORIENT. CXR|PV (CXR subset) and WAVE DEPTH tasks. Tab. 3 shows minimal performance variations. On

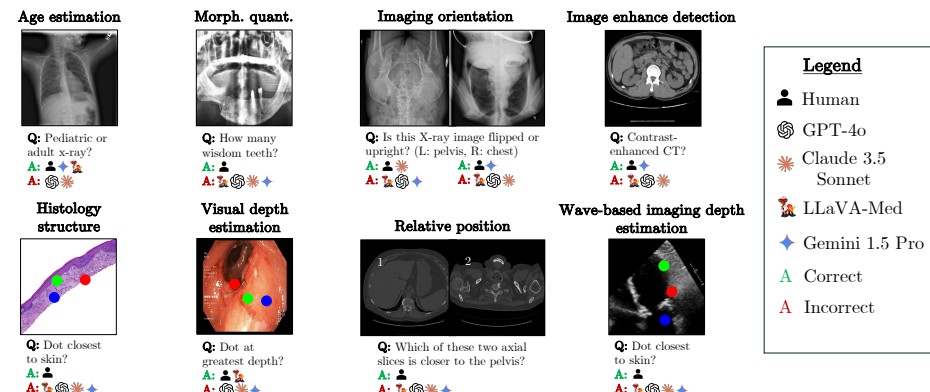

Figure 2: Evaluation of multimodal LLMs on MEDBLINK tasks. Performance is compared between CLAUDE 3.5 SONNET, GEMINI 1.5 PRO, GPT-4O, LLAVA-MED, with human accuracy as a reference.

Table 2: Accuracy (%) of spatial reasoning models on MEDBLINK tasks. Task names are abbreviated with test set size in parentheses. Best-performing model per task is underlined.

| | HISTO. ST. (141) | WAVE DEPTH (146) | VIS. DEPTH (144) | Average (431) |
|---|---|---|---|---|
| Random Choice | 33.3 | 33.3 | 33.3 | 33.3 |
| Experts | **99.2** | **100.0** | **95.1** | **98.1** |
| AURORA (Bigverdi et al., 2024) | 34.7 | 41.7 | 22.5 | 32.9 |
| LLaVA 1.5 7B (Liu et al., 2023) | 34.0 | 31.5 | 38.2 | 34.6 |
| Spatial RGPT (Cheng et al., 2024) | 46.5 | 39.0 | 38.3 | 41.3 |
| GEMINI 1.5 PRO (Team et al., 2024) | 64.0 | 60.6 | 76.5 | 67.0 |

ORIENT. CXR|PV, all methods achieved identical 50% accuracy, suggesting negligible prompting impact. For WAVE DEPTH, FF achieved highest performance (35.6%), a modest 1.4% improvement over MC baseline (34.2%), while OQA and OPB underperformed at 33.6%.

In addition, we evaluated few-shot prompting on two models: CLAUDE 3.5 SONNET and LLAVA-ONEVISION (Tab. 1). Results revealed strong model-dependent sensitivity: CLAUDE 3.5 SONNET improved by 10.4 percentage points with few-shot prompting, while LLAVA-ONEVISION saw an 8.6 point drop. This divergent behavior suggests that few-shot effectiveness varies widely across architectures and cannot be assumed to generalize.

Table 3: Performance comparison across different prompting strategies on medical vision-language tasks. All results reported as accuracy percentages, tested on LLAVA-MED++.

| Prompting Strategy | ORIENT. CXR|PV (CXR) | WAVE DEPTH | Avg. Performance |
|---|---|---|---|
| Multiple-choice (MC) | 50.0% | 34.2% | 42.1% |
| Freeform (FF) | 50.0% | 35.6% | 42.8% |
| OmnimedVQA QA (OQA) | 50.0% | 33.6% | 41.8% |
| Prefix-based (OPB) | 50.0% | 33.6% | 41.8% |

**Is Orientation Perception Harder in Medical Images than Natural Ones?** Results in Tab. 4 suggest MLMs struggle with perception on medical images. For this experiment, we randomly sample 200 images from ImageNet from the following classes: *siberian husky*, *abaya*, *lab coat*, *dining table*, *moving van*, *soap dispenser*, and randomly flip 50% of the images similar to the image orientation task. We then evaluated GPT-4O on predicting whether each image was correctly oriented. On natural images, GPT-4O performed nearly perfectly, making only two errors. On medical images, while the model matched human-level performance on correctly oriented scans, its accuracy dropped substantially to 36% on flipped pelvic X-rays. This disparity suggests that despite strong general perceptual capabilities, MLMs struggle to generalize orientation understanding to the medical domain, revealing a brittleness in their medical visual perception.

**Does Scaling or Diversifying MEDBLINK Yield New Insights?** MEDBLINK complements diagnostic reasoning by focusing on prerequisite perception tasks. Small curated examples (134

Table 4: GPT-4O accuracy on correctly-oriented and disoriented natural and medical images.

| Image Type | Correct | Incorrect Orient. |
|---|---|---|
| Natural | 98.0 | 98.0 |
| Medical (Pelvic Xray) | 100 | 36.0 |

contrast cases) consistently reveal model failures, making scaling unnecessary for new insights. Nonetheless, we verify this to be true by expanding tasks, and retesting with CLAUDE 3.5 SONNET: EST. AGE from 200 (89%) to 1000 (88.1%) samples and WAVE DEPTH from 146 (38.2%) to 963 (41.3%) samples. These results confirm that scaling does not affect the performance. On source diversity, we evaluated the effects of changing the underlying image data source on performance. We tested on the ORIENT. CXR|PV (CXR) task utilizing a different CXR dataset: ChexPert (Irvin et al., 2019) dataset. The results: 86% with CLAUDE 3.5 SONNET vs 82.4% on original, showed negligible variation.

Together, these findings indicate that neither increasing dataset scale nor diversifying image sources provides additional insight into the perceptual limitations of current models.

**Can Models Detect and Reason About Visual Prompts in Medical Images?** Tasks 2,3, and 4 in MEDBLINK require visual prompting. Building on BLINK's findings regarding the influence of color and size in natural images (Fu et al., 2024), we investigate whether models can accurately detect the number and spatial positioning of visual prompts in medical images (Tab. 5). Models demonstrate near-perfect accuracy in basic visual prompt detection (100% on Task 3, 99% on Task 2). For vertical positioning, accuracy ranges from 94.5% to 95.1%, and further improves to 96.3–97.1% when the colored points are spaced at least 10 pixels apart—making detection easier due to clearer separation. However, horizontal position detection reveals a modality-specific gap: while Task 3 (ultrasound) maintains high accuracy (93.1% overall, 95.1% with >10px separation), performance on Task 2 (endoscopy) drops significantly (81.9% overall, 87.1% with >10px). This suggests that endoscopy's visually complex environment poses greater challenges for spatial reasoning compared to ultrasound's simpler grayscale structure. Overall, these results indicate that while models can reliably detect the presence and position of visual prompts, they struggle to interpret their clinical meaning within medical images.

Finally, we tested CLAUDE 3.5 SONNET for visual prompt color-location bias under all color-location (red, blue, green) permutations, on the WAVE DEPTH task. The results for all 6 permutations: P1: 43.1%, P2: 43.8%, P3: 49.3%, P4: 41.8%, P5: 32.8%, P6: 41.4%, with an average of 42.0%, shows a lack of position bias.

Table 5: Accuracy of GPT-4O on medical visual-prompt tasks. *>10 px* denotes point spacing >10 pixels. Task2: wave-based depth; Task3: visual depth.

| Prompt | Task 3 | | Task 2 | |
|---|---|---|---|---|
| | All | >10 px | All | >10 px |
| How many colored circular markers are visible? | 100.0 | – | 99.0 | – |
| Which colored point is positioned highest? | 94.5 | 96.3 | 95.1 | 97.1 |
| Which colored point is furthest to the left? | 93.1 | 95.1 | 81.9 | 87.1 |

**Does Image Resolution Impact Performance on MEDBLINK?** We perform additional resolution experiments on two tasks that should benefit the most from increased resolution: A) HISTO. ST. and B) VIS. DEPTH using GPT-4O, as we can choose between high and low image resolutions per API call. The result: A) Low: 40.0%, High: 37.9%, and B) Low: 68%, High: 73.6% show that resolution has negligible effects on performance on these tasks.

## 5 IMPLICATIONS

Our findings with MEDBLINK have direct implications for the design and evaluation of MLMs in medicine. Current models including leading generalist and domain-specialized systems perform far below human levels on perceptual tasks that clinicians solve effortlessly (best: 76.3% vs. 96.4%). This gap shows that many models lack fundamental visual grounding and therefore cannot yet be trusted for clinical use. Improving perceptual robustness such as depth estimation, counting, and anatomical recognition is essential before deploying these systems in high-stakes settings. MEDBLINK provides a focused benchmark to expose these shortcomings and guide the development of models that meet both clinical performance needs and trustworthiness expectations.

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

# A  MEDBLINK CURATION

## A.1  PROMPT DETAILS: TEXT AND PROMPT

We leverage two main types of prompts: text questions and visual prompts. The questions used for each tasks is outlined in Tab. 9. We use circles/points/dots for visual prompting on 3 tasks. Specifically for both Depth Estimation tasks we use 3 (red, green, blue) colored circles with 10px radius on 512x512 resized image (original 112x112). For the Histology Structure tasks we leverage points whose size depend on the size of the WSI before cropping, specifically we set the size of the circles to 1/70 the size of the max(width, height) of the WSI.

## A.2  HUMAN EVALUATION METHOD

We obtain human evaluation scores from a pool of 4 human experts (3 co-authors, 1 independent). Each task is evaluated by at least one expert and the average score is used as the human benchmark.

## A.3  BENCHMARK STATISTICS

MEDBLINK statistics can be found in Tab. 6, we also outline the Label distribution and count of individual tasks in Table 7

Table 6: Detailed statistics of the MEDBLINK benchmark.

| Statistics | Number |
|---|---|
| Total Questions | 1429 |
| Total Images | 1605 |
| Questions with Visual Prompts | 431 |
| Questions with Multiple Images (2) | 176 |

Table 7: Distribution of labels across different medical imaging tasks.

| Task | Label Distribution | Total |
|---|---|---|
| Task 1 | {yes: 67, no: 67} | 134 |
| Task 2 | {red: 55, green: 38, blue: 51} | 144 |
| Task 3 | {red: 49, green: 51, blue: 46} | 146 |
| Task 4 | {red: 47, green: 53, blue: 41} | 141 |
| Task 5 | {flip: 100, correct: 100} | 200 |
| Task 6 | {1: 127, 2: 49} | 176 |
| Task 7 | {0: 30, 2: 28, 4: 30} | 88 |
| Task 8 | {pediatric: 100, adult: 100} | 200 |

# B  BASELINE MODEL DETAILS

We test 20 Multimodal LMs on MEDBLINK, setting the temperature of all models to 0, including:

1. GPT-5 (OpenAI, 2025)
2. GPT-4O (Hurst et al., 2024)
3. CLAUDE 3.5 SONNET (cla, 2024)
4. GEMINI 1.5 PRO (Team et al., 2024)
5. QWEN 2.5 VL (Bai et al., 2025), specifically, we leverage the 3B, and 7B parameterized models.
6. LLAVA-ONEVISION (Li et al., 2024b). Here we use two versions as well, the 0.5B parameterized model, and 7B parameterized models
7. LLAVA-MED (Li et al., 2023)

8. LLAVA-MED++ (Xie et al., 2024)

9. MED-FLAMINGO (Moor et al., 2023), unlike other models for MED-FLAMINGO to produce valid responses, we need to use few-shot prompting (Sepehri et al., 2024). Specifically, we prompt it with three questions and answers from PMC-VQA benchmark (Zhang et al., 2023) as seen in Tab. 8 for free-from evaluation following MediConfusion (Sepehri et al., 2024) setup.

10. RADFM (Wu et al., 2023b)

11. AURORA (Bigverdi et al., 2024)

12. SpatialRGPT (Cheng et al., 2024)

13. LLaVA 1.5 (7B) (Liu et al., 2023)

14. INTERNVL 2.5 (Chen et al., 2024a), we leverage the 4B, 8B, 26B and 38B parameterized models.

15. LLAMA 3.2 11B (Grattafiori et al., 2024)

## B.1 SMALL SPECIALIZED MODELS

We train small specialized models for some of the tasks with sizeable train sets from the original dataset used to construct the task. We finetune a ResNet-50 (He et al., 2015) model on both the age estimation and image orientation tasks. For training we used a batch-szie of 32, using an 80/20 split we trained each model for 10 epochs and used a learning rate of 1e-3 and decay of 1e-4.

Table 8: Medical imaging prompt template for Med-Flamingo model.

| Model | Prompt |
|---|---|
| Med-Flamingo | You are a helpful medical assistant. You are being provided with images, a question about each image and an answer. Follow the examples and answer the last question. <image>Question: What radiological technique was used to confirm the diagnosis? Answer: Mammography<\|endofchunk\|> <image>Question: What did the CT scan show? Answer: Cerebral edema<\|endofchunk\|> <image>Question: What is the purpose of the asterisk shown in the figure? Answer: To indicate the normal lentoid shape of hypocotyl nuclei.<\|endofchunk\|> <image>Question: **QUESTION** Answer: |

Table 9: Medical imaging tasks with corresponding question formats.

| Task | Question Format |
|---|---|
| Task 1: Image Enhancement Detection | Is this CT scan image contrast-enhanced? (Answer with yes or no) |
| Task 2: Visual Depth Estimation | Which of the dots is at the greatest depth in this endoscopy image? (Answer with red or green or blue) Please put your final answer in 'boxed{}' |
| Task 3: Wave-Based Imaging Depth Estimation | Which of the dots is closest to the surface of the skin? (Answer with red, green, or blue) |
| Task 4: Histology structure | Given this melanoma biopsy, which of the dots is closest in distance to the surface of the skin (stratum corneum or epithelium)? (Answer with red or green or blue) |
| Task 5: Imaging Orientation | Is this X-ray image in its correct anatomical orientation or upside down? (Answer with 'correct' if it is properly oriented, or 'upside down' if it has been rotated 180 degrees.) |
| Task 6: Relative Position | which of these two axial slices is closer to the pelvis, 1 or 2? Please put your final answer in 'boxed{}' |
| Task 7: Morphology Quantification | How many wisdom teeth can be seen in this X-ray image? (A) 0 (B) 2 (C) 4 |
| Task 8: Age Estimation | Is this a pediatric or adult chest x-ray? (Answer with pediatric or adult) |

# C QUALITATIVE EVALUATION OF FAILURE CASES

In the following section, we present case-based qualitative analysis of failure cases to better understand the pattern of failed prediction, Fig. 3, 5, 4, 6, 7, 8, 9, 10, 11, 12.

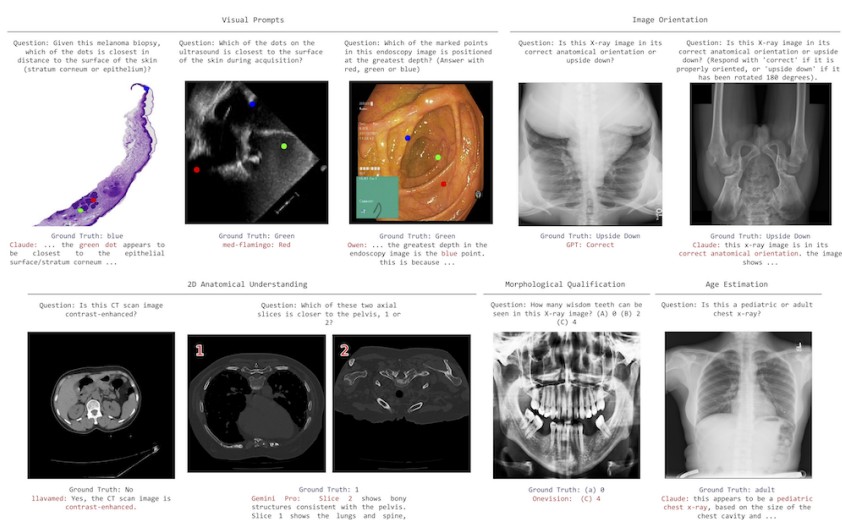

Figure 3: **Examples of *Failure* Cases in the MedBLINK Benchmark.** Examples of failure cases where LLMs struggle with MedBLINK tasks. Incorrect model responses are shown in red, highlighting the challenges LLMs face in handling these fundamental medical perception tasks.

Which of the dots is at the greatest depth in this endoscopy image?

| | |
|---|---|
| Claude | looking at the endoscopic image of what appears to be the colon, the red dot appears to be positioned at the greatest depth. this can be determined by observing the concentric folds of the intestinal wall and how they create a tunnel-like perspective, with the red dot being placed furthest along this tunnel in the deepest visible fold. |
| GPT | the red point is positioned at the greatest depth. |
| Gemini-pro | the red dot is furthest along the visible tract in the image. depth perception in endoscopy images can be tricky, but the folds of tissue and the way the light falls suggest the red dot is deeper within the body |
| Qwen-VL | the point that appears to be positioned at the greatest depth in the endoscopy image is the blue point. this is because it is located further away from the camera and closer to the center of the image compared to the other two points, which are closer to the surface of the tissue. |

Ground Truth: Green

Figure 4: **Example Failure Case in Visual Depth Estimation** Some models incorrectly estimate the relative depth in endoscopic images.

Is this X-ray image in its correct anatomical orientation or upside down? (Respond with 'correct' if it is properly oriented, or 'upside down' if it has been rotated 180 degrees.

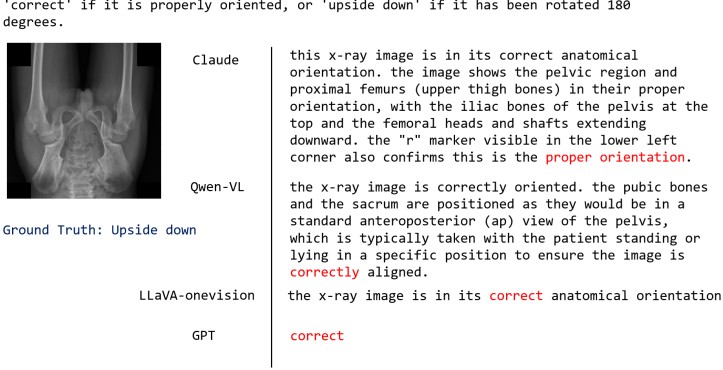

| | |
|---|---|
| Claude | this x-ray image is in its correct anatomical orientation. the image shows the pelvic region and proximal femurs (upper thigh bones) in their proper orientation, with the iliac bones of the pelvis at the top and the femoral heads and shafts extending downward. the "r" marker visible in the lower left corner also confirms this is the proper orientation. |
| Qwen-VL | the x-ray image is correctly oriented. the pubic bones and the sacrum are positioned as they would be in a standard anteroposterior (ap) view of the pelvis, which is typically taken with the patient standing or lying in a specific position to ensure the image is correctly aligned. |
| LLaVA-onevision | the x-ray image is in its correct anatomical orientation. |
| GPT | correct |

Ground Truth: Upside down

Figure 5: **Example Failure Case in Anatomical Orientation Reasoning of MLM on pelvic X-Ray Images** Some models incorrectly classify upside-down pelvic X-Ray images as correct.

Is this a pediatric or adult chest x-ray?

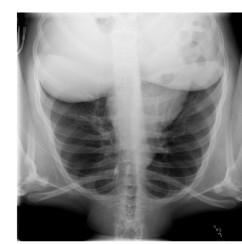

Claude — this appears to be a pediatric chest x-ray, based on the size of the chest cavity and the characteristic appearance of pediatric chest anatomy. the heart size relative to the chest wall and the rib spacing are typical of what you would see in a pediatric patient.

GPT — Pediatric

Ground Truth: Adult

LLaVA-Med — Pediatric

Figure 6: **Example Failure Case in Age Estimation of MLM on Chest X-Ray Images** Some models incorrectly classify adult chest X-Ray images as pediatric and vice versa.

Question: Is this X-ray image in its correct anatomical orientation or upside down? (Answer with 'correct' if it is properly oriented, or 'upside down' if it has been rotated 180 degrees.)

Claude — This chest X-ray is in its correct anatomical orientation. We can tell this because:
1. The heart apex points to the left (as it should)
2. The diaphragm curves upward at the bottom
3. The breast tissue shadows are at the top of the image
4. The vertebral column alignment is correct (superior to inferior)

Ground Truth: Upside Down

Figure 7: **Example Failure Case in Anatomical Orientation Reasoning of MLM on Chest X-Ray Images** Some models incorrectly classify upside-down chest X-ray images as upright, and vice versa.

Question: 'Which of the dots is closest to the surface of the skin? (Answer with red, green, or blue)

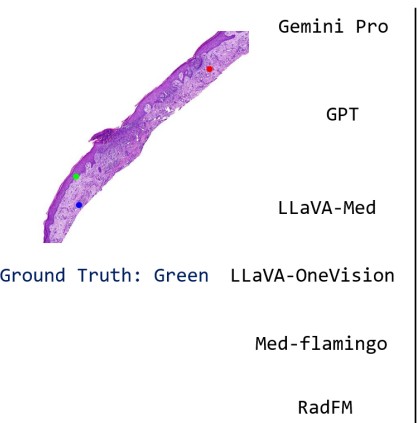

Gemini Pro — Red. The ultrasound probe is placed on the skin, and the red dot is closest to the edge of the image, which corresponds to the probe's location.

GPT — The red dot is closest in distance to the surface of the skin (stratum corneum or epithelium).

LLaVA-Med — The dot closest in distance to the surface of the skin (stratum corneum or epithelium) is the red dot.

Ground Truth: Green    LLaVA-OneVision — Blue

Med-flamingo — Blue

RadFM — The deeper melanoma cells are found in the blue dot.

Figure 8: **Example Failure Case in Distance Reasoning of MLM on pathology Images** Some models incorrectly identify one color to be the closest to the surface of skin, regardless of their location.

Question: Which of the dots on the ultrasound is closest to the surface of the skin during acquisition? (Answer with red or green)

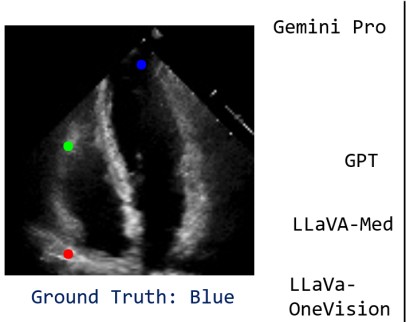

Ground Truth: Blue

Gemini Pro | Red. The image is an apical 4-chamber view of the heart taken with an ultrasound probe placed on the chest. The red dot is closest to the top of the image, which corresponds to the chest wall and skin surface.

GPT | The red dot is closest to the surface of the skin.

LLaVA-Med | The red dot is closest to the surface of the skin.

LLaVa-OneVision | Red

Qwen-VL | The dot that is closest to the surface of the skin is the green dot.

Figure 9: **Example Failure Case in Depth Reasoning of MLM on ultrasound Images** Similar to cases in Fig. 8, models incorrectly identify one color to be the closest to the probe, regardless of their location.

which of these two axial slices is closer to the pelvis, 1 or 2?

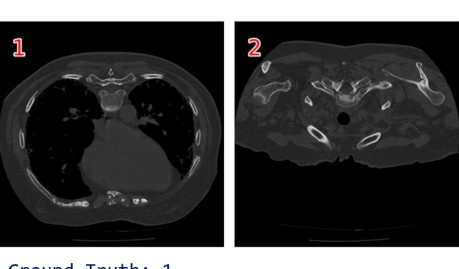

Ground Truth: 1

Gemini-pro | Slice 2 shows bony structures consistent with the pelvis. Slice 1 shows the lungs and spine, which are superior to the pelvis.

GPT | 2

Claude | Looking at these two CT scan axial slices:

Image 1 shows a chest CT scan, as evidenced by the clear visualization of:
- Both lungs (the dark/black areas)
- Heart in the middle (the large white structure)
- Thoracic spine and ribs

Image 2 shows a pelvic CT scan, as evidenced by:
- The characteristic pelvic bone structure
- Hip joints
- Absence of lung tissue

Therefore, 2 is closer to the pelvis.

Figure 10: **Example Failure Case in Relative Position of MLM on axial CT slices** Some models incorrectly orient the anatomical location of CT slices, or confuse the location relationship of two CT slices.

```
How many wisdom teeth can be seen in this X-ray image? (A) 0 (B) 2 (C) 4
```

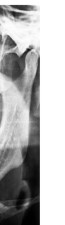

LLaVA-Med | In this X-ray image, there are 4 wisdom teeth visible.

GPT | The X-ray image shows (C) 4 wisdom teeth.

Claude | Looking at this dental panoramic X-ray image, I can see 4 wisdom teeth (option C). The wisdom teeth are visible in all four corners of the jaw - two in the upper jaw (maxilla) and two in the lower jaw (mandible). They appear as the last molars in each quadrant of the mouth. This is a typical presentation of wisdom teeth in a dental X-ray, though their position and alignment can vary between individuals.

Ground Truth: (A) 0

Figure 11: **Example Failure Case in Morphology Quantification** Some models fail to count the number of wisdom teeth, which are distinguishable morphological features.

```
Is this CT scan image contrast-enhanced? (Answer with yes or no)
```

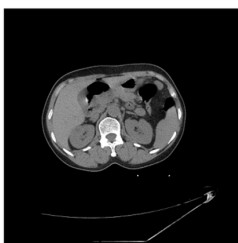

Claude | Yes, this is a contrast-enhanced CT scan, as evidenced by the bright (white) appearance of blood vessels and certain structures in the abdomen. The vertebral body (spine) also shows the typical bright appearance of contrast in the surrounding vessels.

Ground Truth: No

Figure 12: **Example Failure Case in Image Enhancement Detection** Some models fail to detect enhanced regions and misinterpret whether the CT slice is contrast-enhanced.

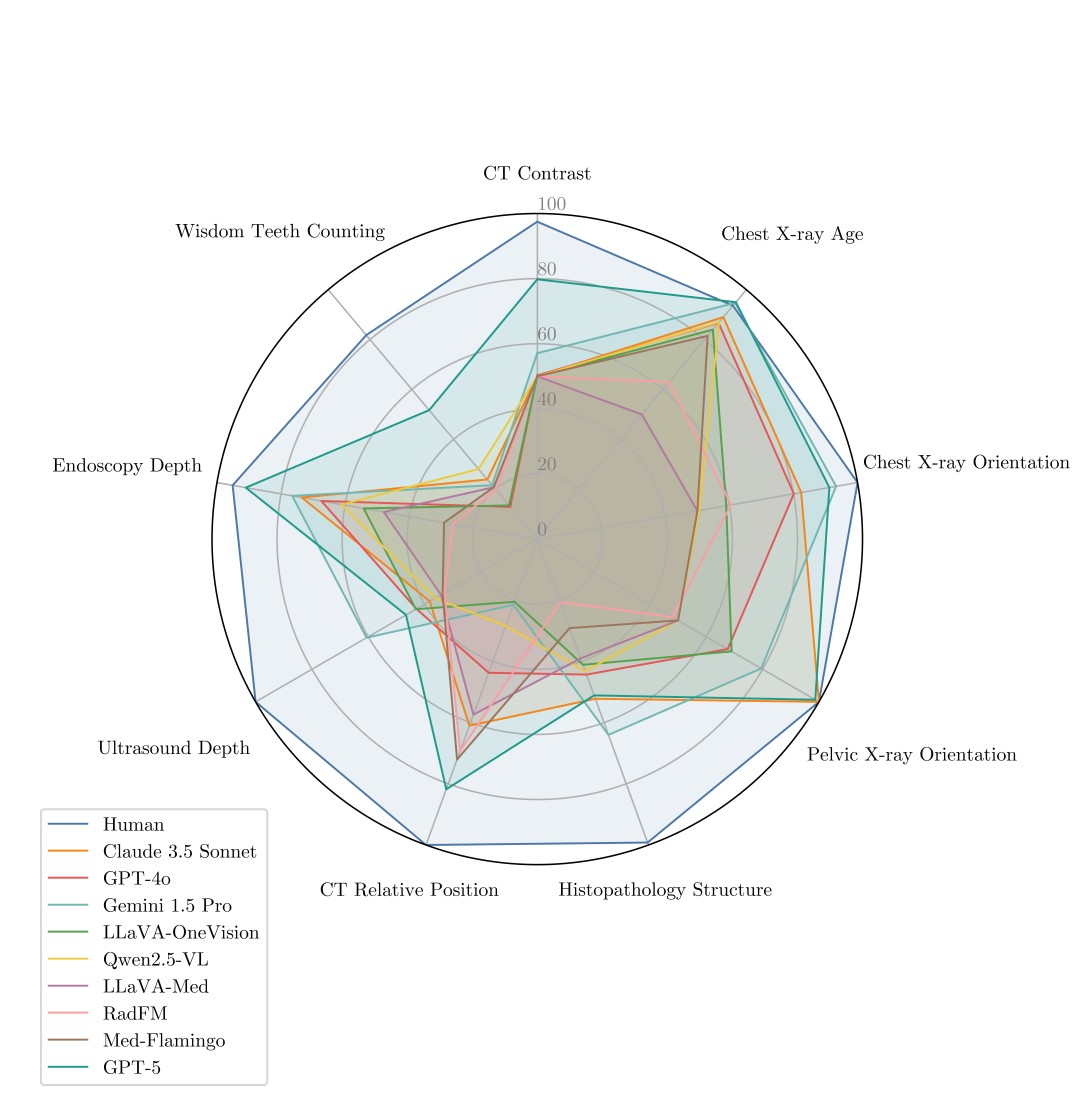

Figure 13: Accuracies of multimodal LMs on MEDBLINK.

