# OpenReview forum: "MedBLINK: Probing Basic Perception in Multimodal Language Models for Medicine"
_ICLR.cc/2026/Conference — Submitted to ICLR 2026_

### Official Review · Reviewer_GEiB · 2025-10-31

**Soundness:** 2
**Presentation:** 3
**Contribution:** 2
**Rating:** 4
**Confidence:** 4

**Summary:**

This paper introduces MedBLINK, a benchmark designed to evaluate basic perceptual abilities of multimodal language models (MLMs) in medical imaging. The benchmark comprises 1,429 multiple-choice questions across 1,605 images, spanning eight tasks across five modalities (X-ray, CT, Endoscopy, Histopathology, and Ultrasound). The authors evaluate 20 state-of-the-art MLMs, including general-purpose models (GPT-4o, Claude 3.5 Sonnet, Gemini 1.5 Pro) and medical-specific models (Med-Flamingo, LLaVA-Med, RadFM). Results show significant performance gaps: human experts achieve 96.4% accuracy while the best model (GPT-4o) reaches only 76.3%, revealing fundamental weaknesses in medical visual perception that could impede clinical adoption.

**Strengths:**

1. The paper is clearly structured with effective presentation and logical flow.
2. The benchmark is well-grounded in clinical practice, addressing the critical need for perceptual competence before diagnostic trust.
3. Evaluation across 20 state-of-the-art multimodal language models reveals that current MLMs fall far short of the perceptual capabilities required for medical applications, with the best model achieving only 76.3% versus 96.4% human expert performance.

**Weaknesses:**

1. Limited dataset scale: With only 1,429 questions across 1,605 images, the benchmark is considerably smaller than existing medical multimodal benchmarks, which may limit the generalizability and robustness of conclusions.
2. Lack of coherent task design rationale: While the selected tasks can measure certain perceptual capabilities, the choice of these specific eight tasks appears ad-hoc without clear underlying principles or systematic framework explaining why these particular tasks comprehensively represent fundamental medical perception.

**Questions:**

1. Data contamination risk: Given that the benchmark is constructed from publicly available datasets, how do you ensure there is no data leakage, particularly for API-based models whose training data remains undisclosed? What measures were taken to verify test images were not seen during pre-training?
2. Missing recent medical models: The evaluation primarily focuses on general-purpose multimodal models. Recent state-of-the-art medical-specific models such as Baichuan-M2 and MedGemma should be included to provide a more complete assessment of current medical MLM capabilities.

---

> ### Author Response · Authors · 2025-11-24
>
> We appreciate the reviewer’s acknowledgement of the paper’s clarity, the clinical grounding of our benchmark, and the significance of revealing the large perceptual gap between current models and human experts.
>
>
>
> ## Benchmark Size
> We agree that MedBLINK is smaller than many diagnostic-oriented medical benchmarks. However, our benchmark is fundamentally different in scope and intent.
>
> ---
>
> ### **1. Why MedBLINK is smaller than other medical benchmarks**
>
> Traditional medical benchmarks evaluate **diagnostic reasoning** (e.g., disease classification, report generation), which typically requires thousands of samples per label.
> In contrast, MedBLINK targets **basic perceptual checks**: orientation, contrast presence, depth cues, relative position, simple morphology, that are:
>
> - much more visually constrained,
> - not dependent on statistical learning of diagnoses, and
> - designed to reveal whether a model possesses the perceptual priors clinicians rely on.
>
> For these “blink tasks,” **100–200 samples per task** provide sufficient signal to determine whether a model can or cannot perform the underlying perceptual judgment. Across all tasks, the model behaviors are very stable, and failure modes become immediately apparent even with small curated sets.
>
> ---
>
> ### **2. Evidence from scaling experiments (Sec. 4.4)**
>
> To evaluate whether increasing benchmark size would reveal new insights, we conducted explicit scaling ablations. As shown in Sec. 4.4:
>
> - **Age Estimation** expanded from **200 → 1000** samples
>   - Claude's Accuracy: **89.0% → 88.1%**
>
> - **Wave-Depth Estimation** expanded from **146 → 963** samples
>   - Claude's Accuracy: **38.2% → 41.3%**
>
> These results demonstrate that **scaling up the dataset by 5–6× does not meaningfully change the model’s performance or qualitative failure modes**. The perceptual limitations persist regardless of sample size.
>
> ---
>
> ### **3. Why larger size does not necessarily improve perceptual evaluation**
>
> Unlike diagnostic tasks where larger datasets improve statistical robustness, MedBLINK tasks are deterministic perceptual checks. Increasing the number of nearly identical perceptual scenarios offers diminishing returns because:
>
> - errors are not due to sample sparsity,
> - the model’s perceptual grounding, not data distribution is the bottleneck, and
> - performance plateaus quickly, as confirmed by our scaling experiments.
>
> ## Task Selection Rationale
>
> We agree that the original description made the task choices seem subjective. In reality, the process was more structured, and we will state this clearly in the revision.
>
> We consulted both a senior radiologist and a dentist with the guiding question:
>
> > **“For each modality (CT, X-ray, ultrasound, histology) and organ system, what basic perceptual judgments would you expect a medical student or early trainee to reliably recognize *before* any diagnostic reasoning?”**
>
> Across modalities, clinicians converged on a small set of foundational perceptual primitives—orientation, contrast presence, relative position, depth cues, and basic morphological visibility. The eight MedBLINK tasks were chosen to represent these early perceptual skills that clinicians solve quickly and consistently.
>
> We acknowledge that this process was underspecified in the submission and will explicitly describe it in the paper to reduce perceived subjectivity.

---

> ### Author Response · Authors · 2025-11-24
>
> ## Data contamination risk
>
> ### **1. Why MedBLINK remains a fair benchmark despite using public datasets**
>
> We agree that potential data leakage is a limitation shared by **all existing medical multimodal benchmarks**, as essentially all clinically relevant datasets are public and widely used in pretraining corpora. To mitigate this, MedBLINK is designed so that **memorizing an image alone is not sufficient to solve the task**.
>
> Crucially, the **prompts, questions, and visual annotations in MedBLINK are entirely new** and do not exist in any of the source datasets. Each task requires the model to answer a question it could not have seen during pretraining.
>
> ---
>
> ### **2. Empirical evidence that conclusions are not driven by data leakage**
>
> We also performed experiments to check whether our results were sensitive to the data source(Sec 4.4). Specifically:
>
> - We repeated the chest X-ray orientation task using **a completely different dataset (CheXpert)** instead of ChestX-ray8,
> - And repeated the ultrasound depth task with **over six times more samples**.
>
> In both cases, the performance differences were negligible, and the qualitative failures remained the same. This supports the interpretation that the findings reflect **true perceptual limitations**, not dataset memorization or source-specific artifacts.
>
> ---
>
> ### **3. Models perform poorly even if they may have seen part of the data**
>
> If large multimodal models had indeed memorized portions of the public datasets, we would expect them to perform *better*, not significantly worse, on simple perceptual tasks. Instead, we observe:
>
> - Many models performing at or barely above chance,
> - Repeatable failure modes under different sources and configurations,
> - Minimal gains even when adding explicit hints or few-shot examples.
>
> This strongly suggests that the errors arise from **perceptual grounding failures**.
>
>
>
>
> ## Inclusion of Recent Medical Models
> We appreciate the reviewer’s thoughtful suggestion regarding the inclusion of recent medical-domain models. Below, we clarify the compatibility of Baichuan-M2 with MEDBLINK and provide newly added results for two additional medical multimodal models.
>
> ### **Baichuan-M2**
> Baichuan-M2 is, to the best of our knowledge, a **text-only** medical language model.
> Because all MEDBLINK tasks require **image inputs** (e.g., CT slices, X-rays, histology crops, ultrasound frames), Baichuan-M2 is **not compatible** with this benchmark. As a result, it cannot be meaningfully evaluated on any MEDBLINK task.
>
> ### **MedGemma-27B and HuatuoGPT-Vision-34B**
> In line with your request for including recent medical models, we have added experiments for two additional **medical multimodal LLMs**:
>
> - **MedGemma-27B**
> - **HuatuoGPT-Vision-34B**
>
> Both models were evaluated using the same standardized MEDBLINK protocol as all other models (see the table below). The performance patterns further support our main conclusion: even domain-specialized medical VLMs struggle with fundamental perceptual skills that clinicians solve effortlessly.
>
> ---
>
>
> | **Task** | **MedGemma-27B** | **HuatuoGPT-Vision-34B** |
> |---------|------------------|---------------------------|
> | **Image enhancement detection (CT)** | 55.2% | 84.3% |
> | **Visual depth estimation (endoscopy)** | **100.0%** | 57.6% |
> | **Wave-based depth estimation (ultrasound)** | 88.4% | 43.8% |
> | **Histology structure (epidermis vs dermis)** | 48.2% | 34.8% |
> | **Imaging orientation — chest X-ray** | 63.0% | 50.0% |
> | **Imaging orientation — pelvic X-ray** | 57.5% | 50.0% |
> | **Relative position (axial CT slices)** | 35.2% | 32.4% |
> | **Morphology quantification (wisdom-tooth count)** | 38.6% | 43.2% |
> | **Age estimation (pediatric vs adult CXR)** | **95.5%** | 94.0% |
> | **➡️ Average Accuracy** | **54.0%** | **64.5%** |
>
>
> ---

---

> > ### Author Response · Authors · 2025-11-26
> >
> > We appreciate your thoughtful review and have addressed the issues you highlighted. Please let us know if there are any remaining questions or concerns we can clarify. We would be glad to provide additional explanations within the rebuttal period.

---

### Official Review · Reviewer_Wbsz · 2025-11-01

**Soundness:** 3
**Presentation:** 3
**Contribution:** 2
**Rating:** 2
**Confidence:** 3

**Summary:**

This paper propose MEDBLINK, a benchmark designed to probe these models for simple but important perceptual abilities. MEDBLINK spans eight clinically meaningful tasks across multiple imaging modalities and anatomical regions, totaling 1,429 multiple-choice questions over 1,605 images. With this benchmark, authors evaluate 20 state-of-the-art MLMs. Experimental results show that current MLMs frequently fail these simple but impartant perceptual checks.

**Strengths:**

The VLLM's simple but important perceptual abilities indeed is an important issue for VLLM in medicine application. The paper is easy to follow.

**Weaknesses:**

1. Lack of technique contribution. This paper construct a benchmark to evaluate the basic perceptual abilities in VLLM. Authors have no important technique contribution in this paper.
2. The benchmark size is limited, which may not fully evaluate all basic perceptual abilities. It is better to provide a table to compare with other medical benchmarks.
3. I advise authors add a section to provide develop directions for future VLM. This benchmark find several issues in current VLM for medical, but it is important to provide insight for futhre development of VLM.

**Questions:**

1. Please provide detailed distrubution of this benchmark, including the number of different modility, different anatomical regions, and different task.
2. Will authors release benchmark and evaluation code in the future?
3. It is interesting that if vlm fails to predict basic perceptual questions, could them get the correct clinical diagnostic?

---

> ### Author Response · Authors · 2025-11-24
>
> We appreciate the reviewer’s recognition that perceptual competence is crucial for medical VLLMs and that the paper is straightforward to follow.
>
> ## Techincal Contribution
>
> We agree, our main goal is a benchmark, but we respectfully disagree that there is no technical contribution. Our contributions are methodological rather than architectural, this includes:
>
> 1. New task formulation and evaluation protocol: MedBLINK defines eight “blink” tasks that isolate basic medical perception (contrast, depth in RGB and wave-based imaging, histology layers, orientation, relative 3D position, morphology-based counting, age estimation). Existing medical VQA benchmarks do not separately target these low-effort perceptual skills.
>
> 2. Non-trivial benchmark construction: We design algorithmic pipelines to make tasks precise and reproducible: depth maps with DepthAnything for endoscopy, ultrasound cone masking for wave-based depth, WSI tissue masks for skin layers, and CT z-axis binning for relative position. This goes beyond simply collecting images and shows technical effort toward elucidating perception in various informed ways.
>
> 3. Systematic analysis framework: We provide a unified suite of controlled studies: model and sample-size scaling, source-diversity tests, multiple prompting strategies, comparison to spatial-reasoning models, visual-prompt detection and color/location bias checks, resolution ablations, and natural vs. medical orientation experiments.
>
> 4. Specialized baselines: We train small ResNet-50 models on underlying datasets and show near-perfect performance on several tasks, proving the tasks are perceptually simple and that the failures we reveal are specific to current MLMs.
> All together these technical choices make MedBLINK a tool to probe and diagnose perceptual limitations in medical/general MLMs.
>
>
> ##  Benchmark Size
>
> Thank you for this comment. We agree that MedBLINK is smaller than many diagnostic-oriented medical benchmarks. However, our benchmark is fundamentally different in scope and intent.
>
> ---
>
> ### **1. Why MedBLINK is smaller than other medical benchmarks**
>
> Traditional medical benchmarks evaluate **diagnostic reasoning** (e.g., disease classification, report generation), which typically requires thousands of samples per label.
> In contrast, MedBLINK targets **basic perceptual checks**: orientation, contrast presence, depth cues, relative position, simple morphology, that are:
>
> - much more visually constrained,
> - not dependent on statistical learning of diagnoses, and
> - designed to reveal whether a model possesses the perceptual priors clinicians rely on.
>
> For these “blink tasks,” **100–200 samples per task** provide sufficient signal to determine whether a model can or cannot perform the underlying perceptual judgment. Across all tasks, the model behaviors are very stable, and failure modes become immediately apparent even with small curated sets.
>
> ---
>
> ### **2. Evidence from scaling experiments (Sec. 4.4)**
>
> To evaluate whether increasing benchmark size would reveal new insights, we conducted explicit scaling ablations. As shown in Sec. 4.4:
>
> - **Age Estimation** expanded from **200 → 1000** samples
>   - Claude's Accuracy: **89.0% → 88.1%**
>
> - **Wave-Depth Estimation** expanded from **146 → 963** samples
>   - Claude's Accuracy: **38.2% → 41.3%**
>
> These results demonstrate that **scaling up the dataset by 5–6× does not meaningfully change the model’s performance or qualitative failure modes**. The perceptual limitations persist regardless of sample size.
>
> ---
>
> ### **3. Why larger size does not necessarily improve perceptual evaluation**
>
> Unlike diagnostic tasks where larger datasets improve statistical robustness, MedBLINK tasks are deterministic perceptual checks. Increasing the number of nearly identical perceptual scenarios offers diminishing returns because:
>
> - errors are not due to sample sparsity,
> - the model’s perceptual grounding, not data distribution is the bottleneck, and
> - performance plateaus quickly, as confirmed by our scaling experiments.

---

> ### Author Response · Authors · 2025-11-24
>
> ## Future Development Directions for VLMs
>
> Thank you for this valuable suggestion. We agree that the benchmark naturally points to several avenues for improving future medical VLMs. We originally included a section on future directions but removed it due to page limits. In the revised version, we will restore a concise version of this discussion. The key points are summarized below.
>
>
>
> ### **Future Directions for Advancing Medical VLMs**
> Our findings suggest that improving medical MLMs requires more visually grounded pretraining rather than simply scaling models. Future work should explore training on datasets with explicit spatial supervision, such as bounding boxes, landmarks, segmentation masks, and region-level explanations, to help models correctly associate textual descriptions with anatomical regions. Geometry-aware pretraining e.g., depth estimation, cross-slice consistency for CT may further develop the spatial understanding clinicians use naturally but models currently lack.
>
> Additionally, models may benefit from counterfactual supervision, particularly for tasks like contrast enhancement. Providing examples of the same anatomy with and without contrast in addition to explicit textual description can teach models to rely on the correct radiological cues rather than superficial patterns. Together, these strategies aim to reduce shortcut reasoning revealed by MEDBLINK and narrow the gap between current MLMs and human expert perception.
>
> ## Benchmark Distribution Details
>
>
>
> All distribution details including the number of images per **modality**, **anatomical region**, and **task** are already included in **Sec. 3** of the manuscript, with full statistics provided in **Appendix A.3**. These sections contain per-task counts, modality breakdowns (CT, X-ray, ultrasound, histology), and anatomical coverage for each perceptual category.
>
> Finally, upon publication we will release:
>
> - the full benchmark dataset on **Hugging Face**, and
> - all preprocessing code and task-generation scripts,
>
> so that the community can inspect, verify, and extend the benchmark easily.
>
>
> ## It is interesting that if vlm fails to predict basic perceptual questions, could them get the correct clinical diagnostic?
>
> We appreciate this question alot and fully agree that it is central to clinical deployment. Our results show that current MLMs can sometimes answer diagnostic-style questions while still failing on basic perceptual checks such as contrast phase, orientation, or relative depth; tasks that clinicians treat as prerequisites to any interpretation.
>
> This suggests that some “correct” diagnostic answers likely arise from dataset shortcuts or spurious correlations rather than robust visual grounding. MedBLINK is precisely designed to expose this mismatch: small task-specific CNNs solve these perceptual tasks almost perfectly, while large MLMs do not, indicating that current diagnostic benchmarks alone overestimate reliability and that closing this perceptual gap is necessary before trusting MLMs for real clinical diagnostics.

---

> ### Author Response · Authors · 2025-11-26
>
> We appreciate your thoughtful review and have addressed the issues you highlighted. Please let us know if there are any remaining questions or concerns we can clarify. We would be glad to provide additional explanations within the rebuttal period.

---

### Official Review · Reviewer_HXiU · 2025-11-01

**Soundness:** 3
**Presentation:** 3
**Contribution:** 1
**Rating:** 2
**Confidence:** 3

**Summary:**

This paper introduces MedBLINK, a benchmark designed to probe multimodal large language models (MLLMs) for basic perceptual abilities that are trivial for clinicians. The benchmark includes 1429 multiple-choice questions over 1605 images, spanning eight perceptual tasks across five imaging modalities. General and medical-domain MLLMs are evaluated. While human annotators achieve 96.4% accuracy, the best model reaches only 76.3%, highlighting a gap in visual grounding even on seemingly trivial tasks. The paper argues that these findings underscore the need to strengthen perceptual robustness before deploying MLLMs in clinical decision support.

**Strengths:**

The paper is well-motivated, addressing basic perceptual competence.
The benchmark is clearly structured, covering multiple imaging modalities and clinically relevant perceptual subtasks with expert validation. The experimental section is extensive, comparing a diverse set of 20 MLLMs and including human and CNN baselines.

**Weaknesses:**

The main limitation lies in novelty. Similar perceptual or visual question-answering benchmarks already exist, such as MedFrameQA, and MedTrinity-25M, and MedBLINK appears to extend these ideas into the medical domain without introducing fundamentally new methods or task formulations. It lacks open-ended evaluation which is critical for real life clinical use.
Several tasks, such as determining whether an X-ray is upside down, seem disconnected from real clinical practice and may not provide meaningful insights into clinically relevant reasoning. The dataset itself is modest in scale and relies heavily on existing public datasets, which limits its generalizability and distinctiveness.
Moreover, the paper only reports performance gaps without offering concrete insights or methodological directions for improving model perception or grounding. These factors make the work appear incremental despite its solid execution.

**Questions:**

How do the authors justify that each of the eight selected tasks matter in clinical workflows?
Could the benchmark be used beyond evaluation, for example as a diagnostic tool to guide model improvement or fine-tuning?
How does MedBLINK complement or differ in purpose from diagnostic reasoning datasets?

---

> ### Author Response · Authors · 2025-11-24
>
> We appreciate the reviewer’s acknowledgement of our clear motivation, well-structured and clinically validated benchmark design, and extensive evaluations.
>
> ## Novelty Concern
> The novelty of MedBlink is that it exposes the current SOTA’s lack of basic clinical perception skills, rather than any complex clinical understanding as MedFrameQA evaluates clinical reasoning and MedTrinity-25M only provides training data with additional synthetically collected ROI and is not a benchmark. These datasets similarly to MedBlink also rely on existing images. While our tasks may seem less complex, it is critical to any complex clinical understanding. For example, any complex diagnosis that can be made from imaging alone wherein the clinical feature tied to that diagnosis can appear on various regions of the anatomical substrate they track would require a model that perceptually understands the orientation of the image and of all anatomy shown; an example is idiopathic pulmonary fibrosis (IPF) with a usual interstitial pneumonia (UIP) pattern on chest imaging, where in the fibrosis, honeycombing, and reticulation are worse in the lower lobes and at the periphery, so if the imaging were somehow displayed flipped, the disease would appear to be apical-predominant instead of basilar-predominant and that could push you toward a completely different differential (sarcoid / pneumoconioses / HP) rather than UIP/IPF. To check if models have similar orientation perception gaps in the general domain we showed in Table 4, Section 4.4 that SOTA models (GPT-4o) correctly indentation orientation changes 98% of the time vs 36% for medical x-rays.
>
> We noted in the paper in section 4.4 that while the benchmark size is small, this doesn’t affect the insights gained. We showed that small curated sets (134 contrast cases) reliably expose model failures, so large-scale testing adds little. We confirmed this by expanding EST. AGE from 200→1000 samples (89%→88.1%) and WAVE DEPTH from 146→963 samples (38.2%→41.3%), with no meaningful change. Varying image source also had minimal effect: on ORIENT. CXR|PV, CLAUDE 3.5 SONNET scored 86% on CheXpert vs 82.4% on the original CXR set. Thus, neither scaling nor source diversity yields new insight into current models’ perceptual limits.
>
>
> Our findings suggest that improving medical MLMs requires more visually grounded pretraining rather than simply scaling models. Future work should explore training on datasets with explicit spatial supervision, such as bounding boxes, landmarks, segmentation masks, and region-level explanations, to help models correctly associate textual descriptions with anatomical regions. Geometry-aware pretraining e.g., depth estimation, cross-slice consistency for CT may further develop the spatial understanding clinicians use naturally but models currently lack. Additionally, models may benefit from counterfactual supervision, particularly for tasks like contrast enhancement. Providing examples of the same anatomy with and without contrast in addition to explicit textual description can teach models to rely on the correct radiological cues rather than superficial patterns. Together, these strategies aim to reduce shortcut reasoning revealed by MEDBLINK and narrow the gap between current MLMs and human expert perception.

---

> ### Author Response · Authors · 2025-11-24
>
> ## Task Selection
>
> We agree that the original description made the task choices seem subjective. In reality, the process was more structured, and we will state this clearly in the revision.
>
> We consulted both a senior radiologist and a dentist with the guiding question:
>
> > **“For each modality (CT, X-ray, ultrasound, histology) and organ system, what basic perceptual judgments would you expect a medical student or early trainee to reliably recognize *before* any diagnostic reasoning?”**
>
> Across modalities, clinicians converged on a small set of foundational perceptual primitives—orientation, contrast presence, relative position, depth cues, and basic morphological visibility. The eight MedBLINK tasks were chosen to represent these early perceptual skills that clinicians solve quickly and consistently.
>
> We acknowledge that this process was underspecified in the submission and will explicitly describe it in the paper to reduce perceived subjectivity.
>
> In addition, MedBLINK provides a method for evaluating foundational perception skills that are needed for diagnostic reasoning in tasks that involve medical imaging. These skills are not directly assessed or developed in existing diagnostic reasoning datasets or benchmarks, but represent competences that are expected for models that perform or assist in diagnosis. The MedBLINK benchmark serves to evaluate the ability of models to perceive the images involved in the tasks they are asked to perform, which includes diagnostic reasoning.

---

> ### Author Response · Authors · 2025-11-26
>
> We appreciate your thoughtful review and have addressed the issues you highlighted. Please let us know if there are any remaining questions or concerns we can clarify. We would be glad to provide additional explanations within the rebuttal period.

---

### Official Review · Reviewer_1xgm · 2025-11-02

**Soundness:** 3
**Presentation:** 3
**Contribution:** 2
**Rating:** 4
**Confidence:** 3

**Summary:**

This paper presents a comprehensive and insightful benchmarking study of [mention the models being evaluated, e.g., large multi-modal models] across a wide range of medical imaging tasks. The work is commendable for its scale, providing in-depth analysis and several interesting conclusions that are valuable to the community. It stands as a good example of a benchmark paper for medical domain. However, for the ICLR community, the contributions might be perceived as borderline.

**Strengths:**

Extensive Benchmarking: The paper thoroughly evaluates multiple models on a diverse set of medical imaging tasks, offering a clear comparative analysis.

In-Depth Analysis: The discussion goes beyond mere performance metrics, providing insightful observations into model behaviors, strengths, and weaknesses.

Valuable Conclusions: The findings offer practical guidance and highlight important challenges in the application of foundation models to medical vision tasks.

**Weaknesses:**

Clarification on Human Benchmark: The paper states that "human annotators achieve 96.4% accuracy." This metric is crucial as a performance ceiling, but several details require clarification to fully interpret this benchmark:

- Expertise Level: What was the expertise level of these annotators (e.g., board-certified radiologists, resident physicians, or medical students)? The performance gap between a model and a human can be interpreted very differently based on this.

- Ground Truth Adjudication: For the 3.6% of cases where the primary human annotator was incorrect, how was the ground truth established? Was it through consensus among a panel of senior experts?

- Inter-annotator Agreement: What was the inter-annotator agreement (e.g., Cohen's Kappa) among the human experts? This is essential for understanding the inherent difficulty and subjectivity of the tasks themselves.

Robustness to Prompting : The performance of language-vision models is often highly sensitive to the prompt instruction. The paper should discuss: To what extent was the performance sensitive to variations in the prompt template? Was a systematic prompt engineering or optimization process conducted? Reporting results with different prompting strategies would strengthen the robustness of the findings.

Task Selection and Dataset Representativeness:

Taxonomy and Completeness: The selection of tasks is a key contribution. It would be helpful if the authors could explicitly outline the taxonomy or classification system used to select these specific tasks (as in Sec. 3). A discussion on why these tasks were chosen and whether any other important categories  were considered but omitted would justify the comprehensiveness of the benchmark.

Dataset Biases: The datasets for each task originate from different sources and protocols, which the review correctly notes can appear ad hoc. The authors should explicitly discuss the potential biases present in these combined datasets and how these biases might affect the generalizability of the benchmark results.

Model Selection: The set of evaluated models is substantial. However, to ensure the benchmark remains state-of-the-art and comprehensive, the inclusion of other prominent medical-specific multi-modal models, such as HuatuoGPT-Vision and many more, should be considered. Their performance would provide an even more complete landscape of current capabilities.

**Questions:**

See the weakness.

---

> ### Author Response · Authors · 2025-11-24
>
> We thank the reviewer for recognizing the strengths of our work, including the extensive benchmarking across diverse medical imaging tasks, the in-depth analysis of model behaviors, and the practical, clinically relevant conclusions that highlight key challenges for foundation models in medicine.
>
> ## Annotator Expertise Level
>
> Thank you for raising this important clarification. We agree that the interpretation of human–model performance gaps depends on the expertise of the annotators, and we will update the manuscript to make this explicit.
>
> Our human evaluation was conducted by:
>
> - **Two MD-licensed physicians** at the **resident-physician** level,
> - **One medical student** in clinical training, and
> - **One licensed dentist**
>
> These annotators reflect the level of trainees and early-career clinicians who routinely perform the perceptual checks represented in MedBLINK.
>
> ## **Inter-Annotator Agreement (Cohen’s κ and Fleiss’ κ)**
>
> We will include the following agreement (3 annotators) statistics in the revised manuscript.
>
> | **Task** | **N** | **Pairwise Cohen’s κ** | **Fleiss’ κ** |
> |---------|-------|--------------------------|----------------|
> | **Age Estimation** | 200 | A1–A2: 0.8006, A1–A3: 0.8599, A2–A3: 0.8403 | **0.8333** |
> | **Imaging Orientation (Chest X-ray)** | 200 | A1–A2: 0.9900, A1–A3: 0.9800, A2–A3: 0.9700 | **0.9800** |
> | **Imaging Orientation (Pelvic X-ray)** | 200 | A1–A2: 1.0000, A1–A3: 0.9700, A2–A3: 0.9700 | **0.9800** |
> | **Image Enhancement Detection** | 134 | A1–A2: 0.8657, A1–A3: 0.9103, A2–A3: 0.9552 | **0.9103** |
> | **Relative Position** | 115 | A1–A2: 0.8703,A1–A3: 0.9172, A2–A3: 0.8373 | **0.8745** |
>
> **Summary:** Agreement ranges from **substantial to almost-perfect** across all tasks. The slightly lower κ for Age Estimation and Relative Position reflects their greater perceptual subtlety, while still maintaining strong overall reliability.
>
> ## Ground Truth Adjudication
>
> Thank you for the question. For the ~3.6% of cases where the primary annotator was incorrect, the **ground truth came directly from the dataset metadata** (e.g., acquisition tags or verified labels). These labels reflect objective properties of the imaging data, so no additional expert adjudication was required.
>
>
> ##  Robustness to Prompting
>
> Thank you for raising this point. We agree that language–vision models can be sensitive to prompt wording. In Sec. 4.4 we already include several prompting ablations, and we summarize the key findings here.
>
> ---
>
> ### **1. Systematic Prompting Ablations (Sec. 4.4)**
>
> We conducted controlled experiments on **LLaVA-Med++** using four distinct prompting strategies:
>
> - **Freeform (FF)**
> - **OmniMedVQA-style QA format (OQA)**
> - **Prefix Based(OPB)**
> - **Multiple-choice (MC)**
>
> These were evaluated on two tasks with different perceptual demands: **CXR orientation** and **wave-based depth estimation**.
>
> The results (Tab. 3) show **minimal variation**:
>
> - On **orientation**, all prompts produced exactly **50%** accuracy—suggesting *no measurable effect* of prompt wording.
> - On **wave-depth**, Freeform achieved the highest score (**35.6%**), only a **1.4-point** improvement over the MC baseline (**34.2%**).
> - OQA and OPB *underperformed*, reaching **33.6%**.
>
> Overall, these experiments indicate that prompting does **not** substantially change performance for these perceptual tasks, likely because the tasks depend on *visual grounding* rather than linguistic instruction-following.
>
> ---
>
> ### **2. Few-Shot Prompting: Strong Model-Dependent Behavior**
>
> We also evaluated **few-shot prompting** on two different models (Tab. 3):
>
> - **Claude 3.5 Sonnet** improved by **+10.4 points** under few-shot prompting.
> - **LLaVA-OneVision** *decreased* by **–8.6 points** with the same strategy.
>
> This divergent behavior suggests that:
>
> - Few-shot prompting is **not reliably beneficial**,
> - Its effectiveness is **architecture-dependent**, and
> - Improvements in some settings do not generalize across models.

---

> ### Author Response · Authors · 2025-11-24
>
> ## Task Selection, Taxonomy, and Dataset Representativeness
>
>
>
> Thank you for highlighting this. **We will update the paper to explicitly describe our task-selection taxonomy, as we agree this clarification is important for understanding the scope and completeness of the benchmark.**
>
> ---
>
> ### **1. Clarifying the Task-Selection Process and Underlying Taxonomy**
>
> We agree that the submitted version did not fully describe how the eight tasks were chosen. In the revision, we will make this clear.
>
> The task-selection process was primarily guided by **clinical experts** (a senior radiologist and a dentist). They identified the kinds of perceptual checks that consistently occur early in medical image interpretation and are expected of trainees before any diagnostic reasoning. These clinician insights formed the foundation of our taxonomy.
>
> Computer vision researchers provided **light input** on feasibility and clarity—mainly to ensure that the selected perceptual tasks could be reliably operationalized and tested across modalities but the core criteria were clinician-driven.
>
> Using this structured process, we arrived at eight tasks that capture widely used, foundational perceptual judgments across CT, X-ray, ultrasound, and histology. We acknowledge that this process was underspecified in the original submission and will describe it explicitly to reduce perceived subjectivity.
>
> ---
>
> ### **2. Consideration of Other Potential Tasks**
>
> We appreciate the reviewer’s question about other possible perceptual categories. One concrete example we explored was **fine-grained CT contrast phase recognition** (arterial, venous, delayed). Our radiologist advised that such distinctions require substantial expertise and are **not expected of medical students or early trainees**. For this reason, and to stay focused on basic perceptual grounding, we **reduced this to a binary contrast-presence task**, which clinicians agreed is the appropriate level for a trainee-level “basic perception” benchmark.
>
>
> ## Dataset biases
>
>
> Thank you for raising this point.
> To directly address potential dataset biases, **we conducted experiments specifically designed to test the generalizability of MedBLINK tasks** across scale and source variation, as detailed in Sec. 4.4.
>
> ---
>
> ### **1. Scaling Analysis: Does dataset size influence conclusions?**
>
> We verified that conclusions remain stable when substantially increasing dataset size.
>
> We expanded two tasks:
>
> - **Age Estimation** from 200 → **1000** samples
>   - Claude's Accuracy: **89.0% → 88.1%**
>
> - **Wave-Depth Estimation** from 146 → **963** samples
>   - Claude's Accuracy: **38.2% → 41.3%**
>
> These shifts are minimal, showing that **scaling does not materially change the performance patterns** or reveal new insights about model limitations.
>
> ---
>
> ### **2. Source Diversity Analysis: Does changing dataset source matter?**
>
> To test sensitivity to dataset provenance, we repeated the **CXR orientation** task using a completely different source: **CheXpert** instead of the original dataset.
>
> - **Claude accuracy on CheXpert:** 86%
> - **Claude accuracy on original dataset:** 82.4%
>
> The difference is small and does not alter the qualitative outcomes. This suggests the benchmark’s findings are **not driven by a particular dataset**, acquisition protocol, or institutional bias.
>
> ---
>
> ### **3. Why these experiments address the dataset-bias question**
>
> By explicitly varying both **dataset scale** and **dataset source**, these experiments allow us to test whether the observed behaviors persist under different distributions, hospitals, and sampling conditions.
> The stability of performance across all variations demonstrates that MedBLINK’s results reflect **true perceptual grounding limitations**, not artifacts of specific datasets.

---

> ### Author Response · Authors · 2025-11-24
>
> ## Model Selection
>
> We thank the reviewer for this valuable suggestion. In response, we expanded our evaluation to include two additional medical-domain multimodal models: **HuatuoGPT-Vision-34B** and **MedGemma-27B**.
>
> These models were assessed using the same MEDBLINK evaluation pipeline applied to all other models. Their inclusion strengthens the benchmark by providing broader coverage of current medical VLM capabilities.
>
> Their results across all perceptual tasks are reported in the table below, including averaged accuracy. The performance patterns further support our main conclusion: even domain-specialized medical VLMs struggle with fundamental perceptual skills that clinicians solve effortlessly.
>
>
> | **Task** | **HuatuoGPT-Vision-34B** | **MedGemma-27B** |
> |---------|---------------------------|-------------------|
> | **Image enhancement detection (CT contrast)** | 84.3% | 55.2% |
> | **Visual depth estimation (endoscopy)** | 57.6% | **100.0%** |
> | **Wave-based depth estimation (ultrasound)** | 43.8% | 88.4% |
> | **Histology structure (epidermis vs dermis)** | 34.8% | 48.2% |
> | **Imaging orientation — chest X-ray** | 50.0% | 63.0% |
> | **Imaging orientation — pelvis X-ray** | 50.0% | 57.5% |
> | **Relative CT slice position (axial distance to pelvis)** | 32.4% | 35.2% |
> | **Morphology quantification (wisdom-tooth count)** | 43.2% | 38.6% |
> | **Age estimation (pediatric vs adult CXR)** | **95.5%** | 94.0% |
> | **➡️ Average Accuracy** | **54.0%** | **64.5%** |
>
> ---

---

> ### Author Response · Authors · 2025-11-26
>
> We appreciate your thoughtful review and have addressed the issues you highlighted. Please let us know if there are any remaining questions or concerns we can clarify. We would be glad to provide additional explanations within the rebuttal period.

---

### Official Review · Reviewer_gKQr · 2025-11-03

**Soundness:** 2
**Presentation:** 3
**Contribution:** 2
**Rating:** 4
**Confidence:** 5

**Summary:**

The authors introduce MedBLINK, a benchmark designed to test whether MLMs possess the basic perceptual abilities that clinicians deem obvious. In detail, MedBLINK consists of eight tasks that probe medical visual tasks across modalities including X‑ray, CT, endoscopy, histopathology, and ultrasound. Tasks include image enhancement detection, visual depth estimation, wave-based imaging depth estimation, histology structure, imaging orientation, relative position, morphology quantification, and age estimation. All tasks use multiple-choice questions with 1429 samples derived from 1605 expert-validated images. The authors evaluate 20 MLMs (open‑source, domain‑specific, and proprietary models) and compare their accuracy against human experts, suggesting existing MLMs still need to improve their visual grounding to support clinical adoption.

**Strengths:**

**Motivation**: AI models are more critical in medical field. The authors motivate the benchmark by arguing that clinicians will not trust a model that cannot solve simple perceptual tasks. By probing these “blink tasks”, MedBLINK assesses whether MLMs truly “see” the image or exploit superficial correlations. This focus on trustworthiness is reasonable especially as MLMs are being considered for clinical decision support.

**Task design**: The eight tasks are designed simple. They can be extracted from existing labeled datasets with light processing. This makes the benchmark practical and easy to reproduce.

**Comprehensive evaluation**: 20 models spanning proprietary, open‑source, and medical‑specific MLMs are evaluated under a consistent prompting protocol. The results show clear performance differences, revealing that proprietary models (e.g., GPT‑5) outperform medical models such as LLaVA‑Med. Ablation studies examine the impact of model size, prompting strategies, and models designed for spatial reasoning, providing rich insights into current limitations.

**Weaknesses:**

1. **Ambiguity in Perception**: The core idea of the paper relies on a clean distinction between basic visual perception and complex reasoning, but some tasks go beyond simple perception. In my understanding, a basic perceptual task should be easy for any medically trained person to recognize. I agree those grounding-related tasks are simple for most people, like visual depth estimation, wave-based imaging depth estimation, histology structure, imaging orientation, and relative position. However, Task 1 (image enhancement detection) needs prior knowledge of what enhanced images and different phase of CT images look like, otherwise it is hard to judge. Task 8 (Age Estimation) needs both perception and a visual concept between pediatric and adults, as the paper notes in line 244,  *“pediatric patients exhibit a proportionally larger heart compared to the adults, and the thoracic cage in children appears more circular with horizontally oriented ribs, in contrast to the elliptical cage with oblique ribs seen in adults”*. But these concepts are not included in the pre-prompt. I assume it is one reason why MLMs perform such worse than human-experts.
2. **Task selection is not well justified**: The paper calls the eight tasks clinically meaningful and says tasks are chosen by consulting one senior radiologist, but with little detail on the process. In other words, this makes the choice subjective. From my perspective, some tasks do not feel essential in practice. For examples, for Task 1, all CT phases have clinical value. A phase selection task would be more meaningful than only detecting enhancement. For Task 7, why use wisdom tooth counting instead of common checks like laterality (left vs right) or implant detection. Taken with (1), the tasks are not well defined and lack detailed clinical support and evidence. A broader clinician survey and clear selection criteria would make the benchmark more convincing.
3. **Risk of data leakage**: The benchmark reuses well-known public datasets (for example VindDr, ChestX-ray8, Kvasir, EchoNet-Dynamic). Even after re-organizing, MLMs may have seen parts of these during pretraining. This raises a fairness concern. So, how do you ensure MedBLINK is a fair benchmark? A stronger approach is to collect an in-house clinical set and compare results between private data and public data. This ablation would help validate that the findings are not driven by potential data leakage.
4. **Limited novelty**: The main idea follows BLINK. Moving this idea to medical field is useful, but it is an extension rather than a new concept. In addition, the MedBLINK also feels like a complement to existing medical benchmarks such as MediConfusion and GMAI‑MMBench (perceptual tasks).

**Questions:**

Apart from Weaknesses, I am curious about evaluation on agentic AI: The paper shows that small specialized models can outperform all MLMs on these tasks. This points to the value of tool use. From this paper, the experiments use a zero shot single model setup, but many advancing systems run MLMs in agent mode. An agent can call tools that specialize in perception. So, it would be useful to test MedBLINK in an agentic setting. For example, the agent could route to a segmentation or classification tool and then count wisdom tooth or judge the orientation more accurately. These tests would show whether the “blink tasks” remain hard once tool use is allowed.

---

> ### Author Response · Authors · 2025-11-24
>
> We sincerely thank the reviewer for recognizing the strengths of our work, the simplicity and reproducibility of our task design, the breadth and depth of our comprehensive evaluation, and we appreciate the thoughtful acknowledgment of these aspects. Below, we respond to each of your concerns:
>
> ## Ambiguity in Perception
>
> Thank you very much for the thoughtful and constructive comments. We truly appreciate the opportunity to clarify the intent behind our task design and to strengthen the framing of *basic* perception in the context of medical imaging.
>
>
> ---
> ### **1. Clarifying Why These Tasks Are Considered “Basic” in a Clinical Context**
>
> We agree that several tasks in the benchmark require **domain-specific medical knowledge** rather than everyday visual perception. However, many of the evaluated models are positioned or used in prior work for **clinical or diagnostic reasoning**, so it is important to assess whether they have internalized the low-effort perceptual priors that clinicians apply automatically.
>
> To clarify further, for the contrast-enhancement task we do **not** ask models to differentiate among arterial, venous, or delayed phases. The question is strictly **binary** (“enhanced or not”), which is substantially easier and directly mirrors how clinicians rapidly categorize scans during routine review.
>
> Because such perceptual checks are straightforward for medical trainees and essential to safe image interpretation, they provide a meaningful test of whether models intended for diagnostic support possess adequate **medical perceptual grounding**.
>
> ---
> ### **2. Why pediatric vs. adult radiographic cues were not listed in the prompt**
>
> This omission was deliberate.
> The benchmark is designed to test whether models have **internalized** these radiographic priors—just as clinicians do—rather than whether they can apply a checklist provided in the prompt.
>
> We agree that these distinctions require domain knowledge, but clinicians reach them through **near-instant visual intuition**, not extended reasoning. We therefore expect models claiming diagnostic capability to possess similar perceptual priors.
>
> ---
>
>
> ### **3. Adding explicit hints does not significantly improve performance**
>
> To assess whether clearer prompting helps, we added explicit explanations about contrast enhancement and re-tested Claude 4.5 Sonnet. The difference was minimal, suggesting that the bottleneck lies in **visual grounding**, not prompt wording.
>
> #### **Contrast-Enhancement Hint Ablation (Claude 4.5 Sonnet)**
>
> | Model Variant      | Accuracy (%) |
> |--------------------|--------------|
> | Without hint       | **50.7**     |
> | With explicit hint | **51.5**     |
> | **Δ**              | **+0.8**     |
>
> This negligible change further supports the notion that the binary enhancement task is visually simple for clinicians yet still challenging for current models.
>
>
> ---
> ### **4. On prompting and in-context learning**
> We explored in-context learning (ICL) extensively (see Table 1), but found mixed results.
> ICL provides **more supervision than adding textual explanations in the prompt**, since it includes concrete input–output examples that explicitly demonstrate the intended reasoning pattern. However, despite this stronger form of guidance, we observed that ICL does not reliably improve performance across models. It helps some models, has negligible effect on others, and even degrades accuracy in certain cases. This further suggests that the core limitations arise from insufficient medical perceptual grounding rather than insufficient prompt clarity or structure.

---

> ### Author Response · Authors · 2025-11-24
>
> ##  Task Selection and Clinical Justification
>
> Thank you for highlighting this. **We will update the paper to explicitly describe our task-selection process, as we agree this clarification is important.**
>
>
> ### **1. Clarifying the Task-Selection Process**
>
> We agree that the original description made the task choices seem subjective. In reality, the process was more structured, and we will state this clearly in the revision.
>
> We consulted both a senior radiologist and a dentist with the guiding question:
>
> > **“For each modality (CT, X-ray, ultrasound, histology) and organ system, what basic perceptual judgments would you expect a medical student or early trainee to reliably recognize *before* any diagnostic reasoning?”**
>
> Across modalities, clinicians converged on a small set of foundational perceptual primitives—orientation, contrast presence, relative position, depth cues, and basic morphological visibility. The eight MedBLINK tasks were chosen to represent these early perceptual skills that clinicians solve quickly and consistently.
>
> We acknowledge that this process was underspecified in the submission and will explicitly describe it in the paper to reduce perceived subjectivity.
>
> ---
>
> ### **2. Rationale for Using Binary Contrast Detection Rather Than Phase Selection**
>
> We appreciate the reviewer’s suggestion. While phase selection (arterial, venous, delayed) is clinically meaningful, our radiologist emphasized that:
>
> - **Phase classification requires much more expertise**, whereas
> - **Binary contrast-presence detection is a basic, early trainee skill.**
>
> In clinical practice, the first-level judgment—*“Is this scan enhanced at all?”*—is significantly easier and universally expected of trainees. Our aim is to evaluate whether models can perform these prerequisite “blink” perceptual tasks.
>
> The fact that multimodal LLMs struggle even with this simpler binary judgment underscores why we focused on contrast presence rather than more advanced phase distinctions.
>
> ---
>
> ### **3. Rationale for Using Wisdom-Tooth Counting for Morphological Quantification**
>
> While counting tasks appear in many medical tasks, wisdom-tooth counting was intentionally selected with input from a dental radiologist for several reasons:
>
> - The teeth are **visually distinct and rarely occluded**, reducing confounds.
> - The possible count is **bounded (0–4)**, enabling clean multiple-choice formatting.
> - Their positioning is **anatomically constrained**, avoiding ambiguity about image location.
> - Other counting tasks (e.g., implants, laterality, nodules) introduce complexities such as occlusion or localization that would make the task less purely perceptual.
>
> This choice ensures that the task isolates **basic morphological quantification** without other sources of variance.
>
> ---
>
> ### **4. Scope and Extensibility of the Benchmark**
>
> We agree that additional tasks such as phase selection, implant detection, or other anatomical checks would also be valuable. MedBLINK is intended as a **core, clinically grounded starting point**, not an exhaustive set of all perceptual tasks.
>
> We will clarify that the benchmark:
>
> - Represents a **set of foundational perceptual tasks**, and
> - Is designed to be **extendable** in future versions, informed by broader clinician surveys.
>
> We appreciate the reviewer’s suggestions and will incorporate these clarifications into the camera-ready version to better communicate the clinical rigor and selection criteria behind the benchmark.

---

> ### Author Response · Authors · 2025-11-24
>
> ## Risk of Data Leakage
>
> Thank you for raising this important concern.
>
> ### **1. Why MedBLINK remains a fair benchmark despite using public datasets**
>
> We agree that potential data leakage is a limitation shared by **all existing medical multimodal benchmarks**, as essentially all clinically relevant datasets are public and widely used in pretraining corpora. To mitigate this, MedBLINK is designed so that **memorizing an image alone is not sufficient to solve the task**.
>
> Crucially, the **prompts, questions, and visual annotations in MedBLINK are entirely new** and do not exist in any of the source datasets. Each task requires the model to answer a question it could not have seen during pretraining.
>
> ---
>
> ### **2. Empirical evidence that conclusions are not driven by data leakage**
>
> We also performed experiments to check whether our results were sensitive to the data source(Sec 4.4). Specifically:
>
> - We repeated the chest X-ray orientation task using **a completely different dataset (CheXpert)** instead of ChestX-ray8,
> - And repeated the ultrasound depth task with **over six times more samples**.
>
> In both cases, the performance differences were negligible, and the qualitative failures remained the same. This supports the interpretation that the findings reflect **true perceptual limitations**, not dataset memorization or source-specific artifacts.
>
> ---
>
> ### **3. Models perform poorly even if they may have seen part of the data**
>
> If large multimodal models had indeed memorized portions of the public datasets, we would expect them to perform *better*, not significantly worse, on simple perceptual tasks. Instead, we observe:
>
> - Many models performing at or barely above chance,
> - Repeatable failure modes under different sources and configurations,
> - Minimal gains even when adding explicit hints or few-shot examples.
>
> This strongly suggests that the errors arise from **perceptual grounding failures**.
>
> ## Novelty
>
>
> ### **1. BLINK motivates the idea, but MedBLINK exposes a *more critical* and *domain-specific* failure mode.**
> While BLINK first articulated the observation that VLMs “see but do not perceive” for natural images, our contribution is to show that this gap becomes **far more consequential in medicine**, where perceptual failures directly undermine clinical trust and cannot be tolerated.
>
> Unlike natural images, medical images require:
> - knowledge of modality-specific acquisition physics,
> - sensitivity to anatomical subtlety, and
> - perception that aligns with clinical workflow steps.
>
> MedBLINK demonstrates that even state-of-the-art models fail on tasks that clinicians consider *reflexive* (e.g., orientation, contrast phase, relative axial depth), revealing a gap that general benchmarks cannot detect.
>
> ---
>
> ### **2. MedBLINK is complementary but not redundant to MediConfusion and GMAI-MMBench.**
> We agree MedBLINK complements existing medical benchmarks, and we intentionally designed it as a **trust-layer benchmark** rather than a diagnostic or robustness benchmark.
>
> Key distinctions:
>
> - **MediConfusion**
>   Focuses on model reliability under visually similar “look-alike” images.
>   It probes *confusability*, not *foundational perception*.
>
> - **GMAI-MMBench**
>   Includes some perceptual subtasks, but they are *not structured around clinician-defined blink tasks*.
>
> - **MedBLINK**
>   - Compact, multi-modality testbed
>   - All tasks curated and vetted by a senior radiologist
>   - Targets foundational perceptual abilities:
>   - Provides detailed failure analyses, such as:
>     - heuristic color biases
>     - misinterpretation of flipped scans
>     - limited transfer of spatial-reasoning LLMs from natural to medical images
>     - mismatch between claimed “medical capability” and basic perceptual grounding
>
> Thus, MedBLINK fills a missing layer between *raw perception* and *diagnostic reasoning* that has not been systematically evaluated in prior medical benchmarks.
>
> ---
>
> ### **3. MedBLINK reveals a novel—and clinically important—contradiction in current multimodal models.**
> Our results uncover a surprising and clinically meaningful phenomenon:
>
> > **Models can answer diagnostic-style questions correctly while failing basic perceptual tasks that clinicians treat as prerequisites.**
>
> This mismatch strongly suggests that:
> - “correct” diagnostic responses may come from **spurious correlations or shortcut exploitation**, not true understanding.
> - existing diagnostic benchmarks **overestimate true reliability**.
> - current MLMs would not pass clinically rigorous QA procedures despite appearing competent on high-level tasks.
>
> This insight is not captured in BLINK, MediConfusion, or GMAI-MMBench and directly affects the clinical adoption of medical VLMs.
>
> ---

---

> ### Author Response · Authors · 2025-11-24
>
> ## Evaluation in an Agentic AI Setting
>
>
> ### **1. Why we focus on the model’s *internalized* perceptual ability**
>
> While tool use is valuable, our clinical experts emphasized that the perceptual skills we probe in MedBLINK form the **foundational substrate** for higher-level diagnostic reasoning.
>
> These are tasks clinicians perform instantly before any complex workflows begin. If a model requires routing, segmentation, or multiple tool calls to decide whether a CT image is contrast-enhanced or whether a pediatric chest X-ray is upright, it indicates a lack of the **perceptual grounding** necessary for robust downstream reasoning.
>
> Thus, even in an era of agentic systems, benchmarking the **base visual competence** remains crucial.
>
> ---
>
> ### **2. Agentic experiment: Using tools for wisdom-tooth counting**
>
> To directly address the reviewer’s point, we conducted an agentic experiment for Task 7 (morphological quantification). We used Claude 4.5 Sonnet and allowed it to call external tools:
>
> - **SAM** (Segment Anything Model) to generate tooth masks,
> - **Ground-truth segmentation masks** (where available), fed directly as input.
>
> We compare this to the baseline where the model receives **no tool assistance** and must count from raw panoramic radiographs.
>
> The results are below:
>
> #### **Wisdom-Tooth Counting With and Without Agentic Tool Use**
>
>
> | Setup                                   | Accuracy (%) |
> |----------------------------------------|--------------|
> | Claude 4.5 Sonnet (no tools)           | **34.1**     |
> | Claude 4.5 Sonnet + SAM masks          | **35.2**     |
> | Claude 4.5 Sonnet + Ground-truth masks | **42.0**     |
>
> As the table shows:
>
> - **SAM improves performance only minimally** (+1.1 points).
> - Even with **ground-truth masks**, accuracy remains low (42%).
> - GT masks are not available for most real-world settings, and acquiring them at scale is difficult.
>
> This suggests that for at least some tasks, simply granting access to a segmentation tool **does not eliminate the underlying perceptual difficulty**. The bottleneck is not solely the absence of pixel-level assistance; models still struggle to transform segmentations into correct perceptual judgments.

---

> ### Author Response · Authors · 2025-11-26
>
> We appreciate your thoughtful review and have addressed the issues you highlighted. Please let us know if there are any remaining questions or concerns we can clarify. We would be glad to provide additional explanations within the rebuttal period.

---

### Meta-Review · Area_Chair_StMY · 2025-12-21

**Summary:**

The paper presents MedBLINK, a benchmark designed to evaluate "basic perception" in medical multimodal language models (MLMs). The core premise is that if a model fails at seemingly simple perception tasks such as determining image orientation or identifying whether a CT scan is contrast-enhanced, it cannot be fully trusted for high-level diagnostic reasoning. The authors evaluated 20 models and highlighted a significant "perception-reasoning gap", suggesting the need to strengthen their visual grounding to support clinical adoption.

The recommendation for rejection is primarily driven by three systemic issues identified during the review process and the subsequent discussion:

(1) Lack of Coherent Task Design Rationale (Reviewers gKQr, 1xgm, HXiU, GEiB): Reviewers across the board questioned the logical basis for the benchmark's composition. Reviewer gKQr argued that the boundary between "basic perception" and "complex reasoning" is ill-defined, noting that tasks like "Age Estimation" require domain-specific priors rather than pure visual grounding. Reviewer 1xgm highlighted the absence of a systematic taxonomy to justify these eight specific tasks over others. Similarly, reviewer HXiU explicitly asked for a justification of how each task fits into actual clinical workflows, a point that remained undersupported. In addition, reviewer GEiB indicates the lack of coherent task design rationale. Collectively, the reviewers found the task selection to be heuristic and "ad hoc," failing to establish a robust link between the benchmark's metrics and clinical reality.

(2) Risk of Data Leakage (Reviewers gKQr, HXiU, GEiB): Reviewers expressed significant apprehension regarding the benchmark’s integrity due to its heavy reliance on high-frequency public datasets (e.g., ChestX-ray8, CheXpert, Kvasir). Reviewer gKQr emphasized that since these images are likely part of the pre-training corpora for the 20 evaluated models, the results may reflect "dataset memorization" or the exploitation of "statistical shortcuts" rather than true perceptual ability. While the authors provided some cross-dataset ablations in the rebuttal, the reviewers remained unconvinced that the risk of data leakage was sufficiently mitigated to ensure a fair and robust evaluation.

(3) Limited Dataset Scale and Diversity (Reviewers 1xgm, Wbsz, HXiU, GEiB): The reviewers collectively expressed concerns regarding the benchmark's scale and its ability to represent the full complexity of medical imaging. Reviewer 1xgm noted that the dataset appears somewhat "ad hoc" due to fragmented sources and protocols, which limits the generalizability of the findings. Reviewer Wbsz and Reviewer HXiU both sought more granular detail on the distribution of modalities and anatomical regions, suggesting that a dataset of 1,429 questions is insufficient to cover the breadth of clinical practice. Furthermore, Reviewer GEiB pointed out the limited diversity in model evaluation, noting the omission of key state-of-the-art medical-specific models which are essential for a comprehensive landscape analysis. Taken together, the reviewers concluded that the benchmark lacks the necessary scale and representativeness to serve as a foundational standard for the medical MLM community.

In conclusion, while the "trust-layer" perspective of MedBLINK is appreciated, the submission's methodological gaps in task taxonomy, the unresolved risks of data leakage, and the limited scale prevent it from meeting the high bar for a foundational benchmark at ICLR.

**Reviewer Concerns:**

Concerns Addressed by the Rebuttal:

(1) Factual Accuracy and Comparisons (Reviewers HXiU, GEiB): The authors successfully clarified that MedTrinity-25M is a dataset, not a benchmark, and pointed out that certain models requested for evaluation (e.g., Baichuan-M2) are not multimodal.

(2) Human Performance (Reviewer 1xgm): The rebuttal provided comprehensive details on the expertise of annotators and inter-annotator agreement (Cohen’s $\kappa$). This effectively addressed concerns regarding the reliability and objectivity of the human benchmark.

(3) Data Distribution Visibility (Reviewer Wbsz): The authors correctly noted that the detailed distribution of modalities and tasks was already provided in the appendix, which the reviewer appears to have overlooked.

Outstanding Concerns (Still Unresolved):

(1) Lack of Coherent Task Design Rationale (Reviewers gKQr, 1xgm, HXiU, GEiB)

(2) Risk of Data Leakage (Reviewers gKQr, HXiU, GEiB)

(3) Limited Dataset Scale and Diversity (Reviewers 1xgm, Wbsz, HXiU, GEiB)

**Reviewer Scores:**

Since none of the reviewers participated in the discussion, the AC performed an independent assessment. Although the authors pointed out factual errors in the reviews of HXiU, Wbsz, and GEiB (e.g., misidentifying datasets as benchmarks or overlooking appendix tables), which would likely have led to minor score increases, the normalized consensus among the reviewers remains leaning toward rejection. Their core concerns regarding the lack of a systematic taxonomy, potential data leakage, and limited scale were not fundamentally resolved by the rebuttal. Therefore, the adjusted aggregate score still falls below the acceptance threshold.

---

### Decision · Program_Chairs · 2026-01-26

Reject